# ContactField: Implicit Field Representation for Multi-Person Interaction Geometry

**Hansol Lee**
hansollee@kist.re.kr

**Tackgeun You**
tackgeun.you@kist.re.kr

**Hansoo Park**
hansupark@kist.re.kr

**Woohyeon Shim**
20211362@sungshin.ac.kr

**Sanghyeon Kim**
sangkim98@hanwha.com

**Hwasup Lim**
hslim@kist.re.kr

Center for Artificial Intelligence Research
Korea Institute of Science and Technology
Seoul, South Korea

## Abstract

We introduce a novel implicit field representation tailored for multi-person interaction geometry in 3D spaces, capable of simultaneously reconstructing occupancy, instance identification (ID) tags, and contact fields. Volumetric representation of interacting human bodies presents significant challenges, including inaccurately captured geometries, varying degrees of occlusion, and data scarcity. Existing multi-view methods, which either reconstruct each subject in isolation or merge nearby 3D surfaces into a single unified mesh, often fail to capture the intricate geometry between interacting bodies and exploit on datasets with many views and a small group of people for training. Our approach utilizes an implicit representation for interaction geometry contextualized by a multi-view local-global feature module. This module adeptly aggregates both local and global information from individual views and interacting groups, enabling precise modeling of close physical interactions through dense point retrieval in small areas, supported by the implicit fields. Furthermore, we develop a synthetic dataset encompassing diverse multi-person interaction scenarios to enhance the robustness of our geometry estimation. The experimental results demonstrate the superiority of our method to accurately reconstruct human geometries and ID tags within three-dimensional spaces, outperforming conventional multi-view techniques. Notably, our method facilitates unsupervised estimation of contact points without the need for specific training data on contact supervision.

## 1 Introduction

Accurate 3D representations of multi-person interactions have critical applications in virtual reality, augmented reality, robotics, and surveillance, as human subjects are central to a variety of content and tasks. In particular, modeling interactions involving multiple individuals in close proximity has gathered attention as the modeling of individual humans and simple group activities has matured. However, the precise estimation and reconstruction of 3D human body poses and shapes in close interaction scenarios present significant challenges, mainly due to occlusion, which complicates accurate reconstruction.

The Skinned Multi-Person Linear (SMPL) model [2], one of the most well-known explicit models, has been extensively utilized not only for individual human models [37, 20, 6] but also in multi-

38th Conference on Neural Information Processing Systems (NeurIPS 2024).

person scenarios [8, 4, 39, 41, 40]. However, as an unclothed human model, it struggles to depict local details such as clothing and hairstyles. Additionally, methods using the SMPL model need separate parameter optimization for each person in a scene, which requires intricate coordination when modeling close interactions between individuals. Implicit representations [28, 21, 22] have been researched as alternatives to the SMPL model. Implicit models, with their higher degrees of freedom, are better suited for clearly expressing local details. Successful modeling of various scenes involving individual human avatars and their detailed local features in multi-person scenarios has been achieved in [43, 3, 23]. Nevertheless, the high degrees of freedom inherent to implicit models demand sophisticated neural architecture designs capable of handling multi-view image features and require high-quality data for training.

To address these challenges, we introduce a novel approach that represents multi-person interaction geometries by simultaneously dealing with geometry, identity, and contact fields in scenes without the need for a prior explicit model such as SMPL. Our proposed implicit field is optimized to estimate both the occupancy and identification (ID) fields, distinguishing each person in 3D space and modeling the interaction geometry. This approach enables the consideration of complex interactions between individuals while preserving the spatial information of each individual.

Furthermore, we utilize a multi-view feature transformer [5] and a global scene feature extraction transformer [34, 35, 36] to construct a 3D scene representation, addressing one of the biggest challenges in reconstructing close interactions: dealing with occlusions. By taking into account the global scene features for each point and leveraging latent 3D scene representations and Transformer architecture, we enhance the ability to infer information about occluded parts, which cannot be achieved with standard multi-view images alone. The Transformer, utilizing context provided by positional encoding and 3D scene representations, can infer the structure and position of occluded parts by learning from visible parts and their spatial relations. This capability is crucial for accurately reconstructing models of each person in the scene, even when direct visual information is lacking.

Additionally, we develop a synthetic dataset for multi-person interaction that includes interactions among 2, 3, and 4 individuals, which helps to address more diversity of characters and complex group dynamics in the scene.

Our experiments validate the superiority of our approach over existing methods, demonstrating its capability to accurately reconstruct and assign tag values in 3D space. Our contributions can be summarized as follows:

- We introduce a novel implicit field representation for multiple people in close interaction scenarios that simultaneously estimates multi-person geometries as occupancy fields, ID fields, and contact fields, thereby preserving their spatial relationships and capturing their interactions.

- Our method employs a novel multi-view local-global feature module coupled with a global scene features extraction technique, leveraging latent 3D scene representations to reconstruct individual geometries and assign ID values and contacts in complex, occluded scenarios.

- We demonstrate that our method can reconstruct 3D multi-person figures more effectively than existing methods. Also, we have created a synthetic dataset that models interactions among 2 to 4 individuals to enhance the understanding of group dynamics in close interactions.

The rest of the paper is organized as follows. We review the related works on 3D human representation in Section 2. We explain our method in Section 3 and demonstrate the effectiveness of the method on two datasets in Section 4. We conclude the paper in Section 5.

## 2 Related Work

Reconstructing 3D human models from RGB images or creating human avatars has been a longstanding challenge. An explicit model, the Skinned Multi-Person Linear model (SMPL) [18], dominates the human avatar research by serving as a canonical 3D human model [37, 15, 14, 30, 20, 6]. However, SMPL is an unclothed human model and is limited in its ability to capture local details such as clothing and hairstyles. Hence, implicit representations, including signed distance fields (SDF) [28] and occupancy fields [44], have gathered attention from the community. Pixel-aligned

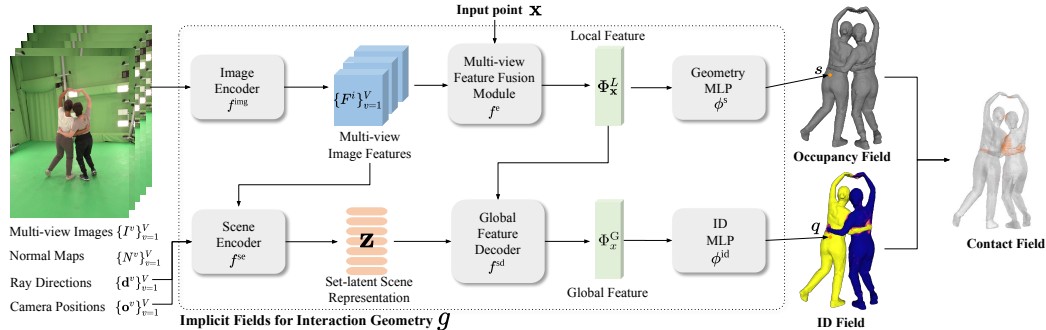

Figure 1: The overall framework of our method. We compute the local and global features from a set of multi-view images and its camera parameters through the proposed multi-view feature local-global transformer. We use local feature to estimate occupancy and global feature to estimate ID at a given point $\mathbf{x}$. From the occupancy and ID fields, we estimate the contact field, as detailed in Section 3.4

feature encoders [32, 33, 13, 1, 16] contextualize the implicit fields by projecting image features onto 3D coordinates using the camera parameters to enhance 3D human avatar reconstruction. Utilizing multi-view RGB images effectively mitigates occluded parts of human in single-view RGB images. DoubleField [38] integrates neural surface and multi-view based radiance fields to represent 3D geometry and the appearance of humans.

Multi-person 3D reconstruction presents unique challenges not encountered in single-person scenarios, such as occlusion, where subjects may obstruct each other's visibility. This task requires not only a detailed understanding of spatial relationships among individuals but also the preservation of individuality while representing their interactions. This process needs precise reconstruction and a deep interpretation of interactions. MPSD [23] employs an implicit approach for the 3D reconstruction of each person, utilizing 6-DOF spatial position estimation within the global scene space. Their method enables multi-person 3D reconstruction from a single image, effectively tracking the locations of individuals. However, because this method primarily addresses data where individuals are spaced widely apart, it does not adequately handle scenarios involving people in close interaction.

Close interaction scenarios [43, 42, 3] are a crucial challenge in multi-person representation. Deep-MultiCap [43] incorporates an attention module and temporal fusion to produce high-fidelity 3D models, relying on the SMPL model for prior segmentation, prediction, and location determination. However, this approach faces challenges, as the accuracy of 3D reconstruction and spatial estimation is bounded by the performance of an initial SMPL prediction, and it requires multiple optimization steps for each individual instance. The work [3] leverages a single-view image and gDNA [7] generative model to generate 3D geometry, refining the spatial positioning of each individual based on contact information. Considering that both methods sequentially reconstruct 3D geometry for each individual, accurately representing multiple subjects in close interaction remains a challenge. Also, existing methods for 3D reconstruction of close human interactions often suffer from data scarcity. The introduction of Hi4D [42] marks a leap forward, offering detailed 4D textures and essential data for studying two-person interactions. However, Hi4D [42] includes only two people, while the dataset described by MultiHuman [43] features up to three. In these two datasets, individuals are captured either separately or together, often overlapping or passing through each other.

Our method innovatively addresses these complexities by employing a transformer-based architecture that integrates multi-view feature fusion with global scene representation, allowing the simultaneous and dynamic reconstruction of multiple interacting individuals. This approach not only captures the detailed geometries of each person but also maintains their unique identities and spatial relationships, even in scenarios with significant occlusions and close physical interactions. Additionally, we created a synthetic dataset that models interactions among 2, 3, and 4 individuals to address even more complex group dynamics.

## 3 Method

Our method introduces a novel representation of multi-person interaction geometry by combining 3D reconstruction with identification of multiple individuals in close interaction from multi-view

images. We achieve this by estimating two key fields: occupancy and ID. The occupancy estimation is crucial for reconstructing the geometry of each person, while the ID estimation facilitates the identification of individuals within the 3D space. We leverage transformer architectures to tackle the significant challenge of occlusions common in close interaction scenarios. Our architecture first extracts local features and then computes global features by integrating these local features with 3D scene representations. This approach ensures robust integration of local and global information across multiple perspectives, enabling a comprehensive understanding and reconstruction of the complex configurations of scenes. Figure 1 represents our process pipeline, encapsulating the essence of our approach.

## 3.1 Implicit Fields for Multi-Person Interaction Geometry

Our method utilizes implicit functions to represent interaction geometry in 3D space. Implicit functions uniquely define the surface of a 3D object by specifying a level set within a field [21]. This representation allows for a continuous definition of the surface, enabling the precise reconstruction of complex geometries. Alongside the geometric reconstruction, we introduce a novel approach to model the ID and contact fields within the same 3D space, providing a method for distinguishing individual entities in closely interacting scenarios.

Our model takes a query point $\mathbf{x} \in \mathbb{R}^3$ to predict two key attributes: the occupancy $s$, and the identification $q$ for each query point. We extract multi-view local and global features, denoted as $\Phi_{\mathbf{x}}(\cdot) = \left[\Phi_{\mathbf{x}}^{\mathrm{L}}, \Phi_{\mathbf{x}}^{\mathrm{G}}\right]$, given a set of multi-view images $\{\mathbf{I}^v\}_{v=1}^V$, their corresponding image normal maps $\{\mathbf{N}^v\}_{v=1}^V$, and camera parameters $\{\mathbf{K}^v\}_{v=1}^V$, where $V$ denotes the total number of views. The proposing model $g(\cdot)$ can be formally defined by the following equation:

$$s, q = g(\mathbf{x}; \Phi_{\mathbf{x}}(\{\mathbf{I}^v, \mathbf{N}^v, \mathbf{K}^v\}_{v=1}^V). \tag{1}$$

The occupancy value $s \in \{0, 1\}$ is a binary indicator signifying whether a point resides inside or outside the surface boundary of an individual, essentially distinguishing the geometric presence of the subject in the 3D space. The identification (ID) value $q \in \mathbb{R}$ provides a unique identifier to each point, allowing differentiation of individuals in close proximity by assigning distinct ID values.

## 3.2 Multi-View Local-Global Feature Module

The architecture of the $\Phi_{\mathbf{x}}$ module, which extracts local and global features $\left[\Phi_{\mathbf{x}}^{\mathrm{L}}, \Phi_{\mathbf{x}}^{\mathrm{G}}\right]$ from multi-view images, determines the quality of occupancy and ID fields. We introduce a local-global feature scheme through a dedicated feature extraction architecture.

Given a set of images $\mathbf{I}^v$, normal maps $\mathbf{N}^v$ generated by the method described in IntegratedPIFu [5], and camera parameters $\mathbf{K}^v$, our method begins by following PIFu [32] using an image encoder [25]. Each image and normal map extracted by [5] is processed individually within the multi-view inputs, allowing distinctive features to be captured from different perspectives. For any given query point $\mathbf{x}$ in 3D space, we project this point onto the 2D planes of the multi-view inputs to acquire pixel-aligned features. Formally, the feature extraction process can be described as follows:

$$F^v = f^{\mathrm{img}}(\mathbf{I}^v \oplus \mathbf{N}^v), \tag{2}$$

$$F_{\mathbf{x}}^v = \Pi(F^v, \mathbf{K}^v, \mathbf{x}), \quad \forall v \in \{1, 2, \dots, V\}, \tag{3}$$

where $F^v$ denotes the set of features extracted from concatenated $v$-th image and $v$-th normal map and $\oplus$ symbolizes channel-wise concatenate operation. $f^{\mathrm{img}}$ represents the image encoder function, designed to process each views.

$F_{\mathbf{x}}^v$ represents the features at the image pixel corresponding to the projection of point $\mathbf{x}$ onto the $v$-th image plane. The function $\Pi(\cdot)$ computes the location on the image plane where the 3D point $\mathbf{x}$ is projected and extracts the relevant features from $F^v$ at that point. For the detailed implementation, refer to PIFu [32].

This approach ensures that the features $F_{\mathbf{x}}^v$ are aligned with the geometry of the scene as observed from multiple viewpoints, facilitating an accurate reconstruction of the 3D space.

The pixel-aligned features extracted from each view are then aggregated through a local-global process to create a comprehensive feature representation for each query point $\mathbf{x}$. This aggregation is

performed using a view-to-view transformer encoder [38], formulated as:

$$\Phi_{\mathbf{x}}^{\mathrm{L}} = f^{\mathrm{e}}(\gamma(\mathbf{x}), F_{\mathbf{x}}^1, F_{\mathbf{x}}^2, \cdots, F_{\mathbf{x}}^V), \tag{4}$$

where $\Phi_{\mathbf{x}}^{\mathrm{L}}$ represents the local feature set obtained by fusing features across all views corresponding to the query point $\mathbf{x}$ with positional encoding $\gamma(\mathbf{x})$.

Inspired by the architectures of the Scene Representation Transformer (SRT) encoder [34] and decoder mechanism [36], the global feature module complements these local features with global scene context. The SRT encoder $f^{\mathrm{se}}$ receives the extracted features from all views along with their corresponding camera positions $\mathbf{o}$ and normalized ray direction $\mathbf{d}$ from each camera information to encapsulate global scene information into a compact representation $\mathbf{z}$. This scene representation serves as an input to the global feature decoder $f^{\mathrm{sd}}$, which extracts global features $\Phi^{\mathrm{G}}$:

$$\mathbf{z} = f^{\mathrm{se}}(\{F^v, \mathbf{o}^v, \mathbf{d}^v\}_{v=1}^V), \tag{5}$$

$$\Phi_x^{\mathrm{G}} = f^{\mathrm{sd}}(\mathbf{z}, \Phi_x^{\mathrm{L}}), \tag{6}$$

where $\Phi^{\mathrm{G}}$ represents the global features for multi-view images.

Finally, we forward the features $\Phi_{\mathbf{x}}(\cdot)$ into two multi-layer perceptrons (MLPs), $\phi^{\mathrm{s}}$ and $\phi^{\mathrm{id}}$ to retrieve occupancy values $s$ and ID values $q$ for each query point:

$$s = \phi^{\mathrm{s}}(\Phi_x^{\mathrm{L}}), \tag{7}$$

$$q = \phi^{\mathrm{id}}(\Phi_x^{\mathrm{G}}). \tag{8}$$

This method ensures a comprehensive feature set that enhances predictions by integrating both localized and globalized scene insights.

Detailed architectures of $f^{\mathrm{e}}$, $f^{\mathrm{se}}$, and $f^{\mathrm{sd}}$ are shown in section A.2 of the Appendix.

## 3.3 Training Objective

We train this model $g$ with following objectives:

$$\mathcal{L} = \omega_s \mathcal{L}_{\mathrm{MSE}} + \omega_{\mathrm{contra}} \mathcal{L}_{\mathrm{contra}} + \omega_{\mathrm{group}} \mathcal{L}_{\mathrm{group}}, \tag{9}$$

where $\mathcal{L}_{\mathrm{MSE}}$ is used to train occupancy field, while $\mathcal{L}_{\mathrm{contra}}$ and $\mathcal{L}_{\mathrm{group}}$ focus on accurately identifying individual entities captured as ID fields.

For occupancy predictions, mean Squared Error (MSE) loss is defined for $N$ query points as:

$$\mathcal{L}_{\mathrm{MSE}} = \frac{1}{N} \sum_{i=1}^{N} (s_i - s_i^{\mathrm{gt}})^2, \tag{10}$$

where $s_i$ is the predicted occupancy and $s_i^{\mathrm{gt}}$ is the ground truth for the $i$-th point. $\mathcal{L}_{\mathrm{MSE}}$ reconstructs 3D geometries by minimizing the discrepancy between the predicted and actual occupancy values, crucial for capturing the intricate details of the scene.

To ensure that points associated with the same object are assigned identical predicted ID values, we tailor an associative embedding [24] consisting of contrastive loss [9, 12] and a grouping loss in the training objective. The contrastive loss is computed based on pairwise Euclidean distances among ID value, considering both positive pairs for the same instance label and negative pairs for a different instance label. For all query points $\{\mathbf{x}_i\}$, we have a set of predicted ID values $\{q_i\}$ and associated ground truth instance labels $\{l(\mathbf{x}_i)\}$ given from datasets. Then, the contrastive loss is formulated as:

$$\mathcal{L}_{\mathrm{contra}} = \frac{\sum_{ij}(m_{ij}^{\mathrm{pos}} \cdot d_{ij})}{P} + \frac{\sum_{ij}\left(m_{ij}^{\mathrm{neg}} \cdot \max(0, -d_{ij} + \delta)\right)}{P}, \tag{11}$$

where $d_{ij} = ||q_i - q_j||$ is a pairwise Euclidean distances between ID values of $i$ and $j$-th query points. $m_{ij}^{\mathrm{pos}} = \mathbb{I}[l_i = l_j]$ and $m_{ij}^{\mathrm{neg}} = \mathbb{I}[l_i \neq l_j]$ are mask value for indicating positive or negative pairs. $\delta$ is a predefined margin threshold, and $P$ is the total number of pairs. $\mathbb{I}[\cdot]$ denotes an indicator function that returns 1 for true and returns 0 for false case. If $\delta = 1$, then the negative loss component is doubled. This function aims to minimize the distance between positive pairs while ensuring negative pairs are separated by at least the margin.

The grouping loss function is devised to refine the embeddings of predicted ID values by accomplishing two primary objectives: minimizing the variance within groups of values that correspond to the same ground truth label and maximizing the separation between groups linked to different labels. This function is articulated as follows:

$$\mathcal{L}_{\text{group}} = \sum_{k=1}^{K} \frac{1}{|G_k|} \sum_{l(\mathbf{x}) \in G_k} (x - \boldsymbol{\mu}_k)^2 + \sum_{k=1}^{K} \sum_{l=1, l \neq k}^{K} e^{-|\boldsymbol{\mu}_k - \boldsymbol{\mu}_l|}. \tag{12}$$

In this equation, $K$ denotes the total number of unique ID values present in the ground truth labels, signifying the distinct classifications for the predicted ID values. Each $G_k$ represents the collection of predicted ID values that match the $k$-th unique ground truth label, forming groups of predictions intended to bear the same ID. The term $|G_k|$ reflects the size of each group, indicating the count of predictions it encompasses. The variable $x$ refers to individually predicted ID value within the group $G_k$, and $\mu_i$ is the average of values in $G_k$, acting as the group's centroid. The notation $|\cdot|$ signifies the absolute value, ensuring that distances in the computations remain non-negative.

The first component of the grouping loss concentrates on reducing the squared distances between each predicted ID value $x$ and its group's mean $\mu_k$, thereby decreasing the variance within each group. Conversely, the second component introduces an exponential penalty on the closeness of centroids $\mu_k$ and $\mu_l$ from distinct groups, fostering clear separation between these groups by penalizing closely situated centroids. This bifocal approach of the loss function is crucial for steering the model towards generating cohesive yet distinctly separated embeddings, which is fundamental for the precise identification and differentiation of unique ID values.

### 3.4 Estimation of Interaction Geometry

To reconstruct the geometry of occupancy fields, we first retrieve the occupancy fields by querying dense points from the implicit fields $g$. After estimating the occupancy field, the geometry is reconstructed using the Marching Cubes algorithm [19]. Initially, we define the bounding box of the voxel grid. The Marching Cubes algorithm is then applied to the grid, where a surface is generated by interpolating the predicted occupancy values with $s > \tau$ within each voxel unit. Subsequently, the algorithm retrieves the ID values from the same voxel units and applies a color map, which facilitates the visual distinction of different individuals represented in the geometry. We use $\tau = 0.5$ for thresholding occupancy values.

The final contact fields can be estimated by the variance of predicted ID values. Close interaction among different instances typically leads to deficient information due to occluded images. Consequently, the uncertainty in the predicted geometry is increased when the information from multiple views is deficient. We choose the variance of predicted ID values among possible options as an uncertainty metric because Euclidean distance is adopted within the training objective for predicted ID values. For each voxel $\mathbf{x} = (x_1, x_2, x_3)$ in the 3D grid $\mathbf{V}$, we extract a local neighborhood $\mathcal{N}(\mathbf{x})$, excluding background value ($s < \tau$). We then calculate the standard deviation of the neighborhood values,

$$\sigma_{\mathbf{x}} = \sqrt{\frac{1}{|\mathcal{N}(\mathbf{x})|} \sum_{\mathbf{v} \in \mathcal{N}(\mathbf{x})} (\mathbf{v} - \boldsymbol{\mu})^2}, \tag{13}$$

where $\mu$ is the mean of the neighborhood values and $|\mathcal{N}(\mathbf{x})|$ is the number of elements in the neighborhood.

A voxel is marked as a contact point if the standard deviation exceeds a threshold of $\tau_c = 0.25$. The formulation is as follows:

$$c(\mathbf{x}) = \begin{cases} \sigma_{\mathbf{x}} & \text{if } \sigma_{\mathbf{x}} > \tau_c \\ 0 & \text{otherwise} \end{cases}. \tag{14}$$

### 3.5 SynMPI: Synthetic Dataset for Multi-Person Interaction

Current multi-human benchmark datasets, such as Hi4D [42] and MultiHuman [43], are limited in size and scope, particularly in terms of the number of interacting individuals and the diversity of interaction scenarios. To address these limitations, we introduce a new synthetic dataset designed

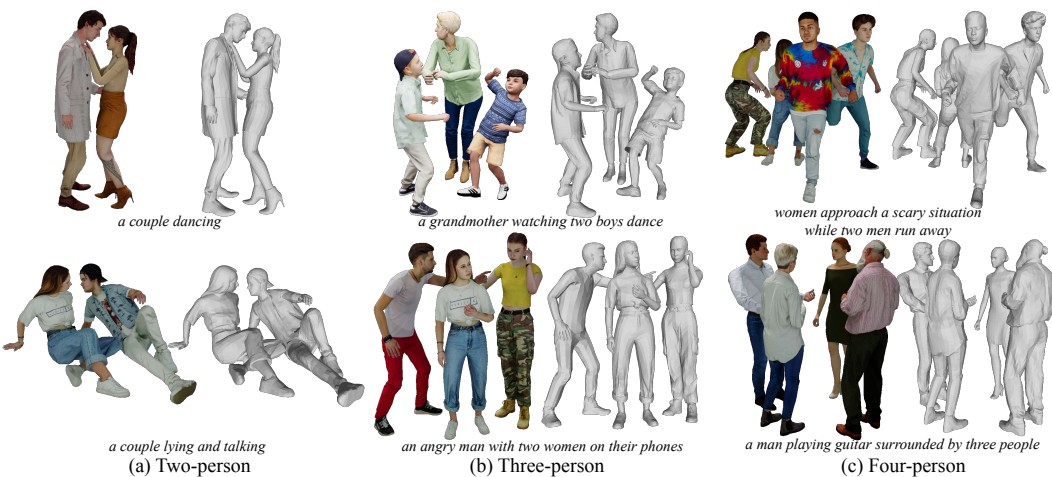

*a couple dancing*     *a grandmother watching two boys dance*     *women approach a scary situation while two men run away*

*a couple lying and talking*     *an angry man with two women on their phones*     *a man playing guitar surrounded by three people*

(a) Two-person     (b) Three-person     (c) Four-person

Figure 2: Examples of multi-person interaction geometry in the SynMPI dataset. Each sample contains interactions involving (a) 2, (b) 3, or (c) 4 people. In each sample, the left side shows rendered RGB images and the right side shows rendered meshes. *Italic sentences* explain the multi-people interaction types of samples adopted from Character Creator 4 [31].

to encompass a broader range of interaction scenarios involving groups of up to four individuals. Figure 2 shows examples from our dataset.

Our dataset is multi-view and supports multi-person interactions, accommodating groups of two to four individuals to effectively capture the spatial dynamics of group interactions. We include elderly individuals and children, representing a wider range of identities for the enhanced diversity of dataset. Additionally, the synthetic data features individuals with varying ages, heights, weights, and clothing styles. We also incorporate multiple types of motions for each participant, enriching the dataset with a diverse set of dynamic interactions. The dataset encompasses approximately 50 distinct motion types, including dancing, running, and talking. The dataset includes 49 characters (26 female, 23 male) with 50 motion sequences across 43 scene configurations, totaling approximately 25,000 frames. For a detailed breakdown of the dataset, please refer to Appendix B.1.

## 4 Experiments

### 4.1 Datasets

We conduct our experiments on Hi4D [42] and the proposed SynMPI dataset, which include various multi-person interactive scenarios. Aggregated training splits of both Hi4D and SynMPI serve as the training set for our experiments. The test splits of each dataset serve as the test set.

**Hi4D [42]** This dataset offers samples which two individuals engage in various interactions, encompassing 20 subject pairs with diverse body shapes and appearances with 8 view images. It includes 3D human scans, instance segmentation masks at the vertices of the 3D scans and image pixels, SMPL [2] parameters, and contact information at the vertex level. We use a random split of 70% for training and 30% for evaluation.

**SynMPI** Our synthetic multi-person interaction dataset captures a wide range of interaction scenarios involving groups of more than two individuals, encompassing a diverse spectrum of dynamic interactions. From the our synthetic datasets, we use images from 8 views and 3D geometry for our experiments. We randomly split samples in SynMPI into 70% for training and 30% for evaluation.

### 4.2 Metrics

Following the evaluation protocols of existing studies [32, 43], we adopt four metrics to assess the quality of the interaction geometry. Chamfer Distance (CD) calculates the bidirectional disparity between points on the predicted and corresponding ground-truth mesh. Point to Surface (P2S) computes the unidirectional distance from each point of the ground-truth mesh to the nearest surface

Table 1: Evaluation results for multi-person interaction geometry. The values presented under the CP row indicate the threshold value, denoted by $\epsilon$, which was employed to construct the pseudo G.T. contact map in 3D space for evaluation.

| Model | Hi4D [42] | | | | | | | SynMPI (Ours) | | |
|---|---|---|---|---|---|---|---|---|---|---|
| | CD↓ | P2S↓ | NC↑ | CP↑ | | | | CD↓ | P2S↓ | NC↑ |
| | | | | 0.025 | 0.05 | 0.075 | 0.1 | | | |
| DMC [43] | 0.631 | 0.495 | 0.768 | - | - | - | - | 0.804 | 0.800 | 0.688 |
| Ours (w/o SRT) | 0.468 | 0.402 | 0.888 | 0.317 | 0.424 | 0.458 | 0.492 | 0.630 | 0.492 | 0.827 |
| Ours | **0.406** | **0.329** | **0.892** | **0.447** | **0.629** | **0.670** | **0.703** | **0.511** | **0.374** | **0.836** |

Table 2: Ablation study on grouping loss function in Eq (12).

| Ablation type | Squared distance | Exponential penalty | CD↓ | P2S↓ | NC↑ | CP↑ | |
|---|---|---|---|---|---|---|---|
| | | | | | | 0.05 | 0.075 |
| (a) | not used | not used | 0.462 | 0.363 | 0.892 | 0.111 | 0.187 |
| (b) | used | not used | 0.400 | 0.314 | 0.892 | 0.345 | 0.528 |
| (c) | not used | used | 0.532 | 0.403 | 0.880 | 0.228 | 0.335 |
| (d) | used | used | 0.406 | 0.329 | 0.892 | 0.629 | 0.670 |

on the corresponding predicted mesh based on the closest-set euclidean distances. Normal Consistency (NC) measures the difference of the normal vector between points on the predicted and corresponding ground-truth mesh with the nearest-set euclidean distances. Contact Precision (CP) is defined by the overlap between the estimated contact map and the pseudo ground truth contact map generated from ground truth meshes. For a detailed definition of the metrics, please refer to the Appendix A.4.

## 4.3 Baseline Models

**DeepMultiCap (DMC) [43]:** We compare our framework with existing methods [43] reconstructing 3D human with multi-view images. DMC leverages a 3D feature of SMPL mesh to infer information of occluded regions during the learning process of the pixel-aligned implicit function. For the integration of features from multiple views, it utilizes a transformer-based approach. They use 8-view images for their reconstruction process, and we followed the same setup. We use the public implementation of DMC [1] and apply LVD [10] on the SynMPI, as well as MVpose [11] on the Hi4D, to obtain SMPL parameters. We refer to Appendix B.2 for additional details.

## 4.4 Results and Analysis

**Quantitative Results** Reconstruction results are evaluated against the baseline method, as shown in Table 1. Our method demonstrated superior performance in terms of reconstruction quality metrics, indicating the effectiveness of our approach in accurately capturing and reconstructing 3D models of multiple people.

**Qualitative Results** Figure 3 illustrates the qualitative performance of our method in generating high-quality reconstructions of multiple people in close interaction scenarios. Compared to DMC [43], our method excels in handling dynamic interactions and heavy occlusion, which are common challenges in multi-person reconstruction tasks. DMC struggles with these scenarios, leading to less accurate SMPL estimations. For additional results, please refer to the appendix and supplementary video.

**Ablation Study on Architecture** We performed ablation studies to assess the impact of our proposed modules. Table 1 presents the performance of the geometry module without the global features. The results demonstrate that global features significantly enhance performance across all geometry-related metrics. Table 1 highlights the impact of global features on contact precision performance, demonstrating enhanced accuracy of contact predictions across a range of thresholds. Figure 5 visually compares contact precision performance with and without global features, illustrating substantial

---

[1] https://github.com/DSaurus/DeepMultiCap.

Table 3: Ablation on the number of views.

| Model | # views | CD↓ | P2S↓ | NC↑ |
|---|---|---|---|---|
| DMC [43] | 4 | 1.304 | 0.922 | 0.705 |
| | 8 | 0.631 | 0.495 | 0.768 |
| Ours | 4 | 0.761 | 0.472 | 0.870 |
| | 8 | 0.406 | 0.329 | 0.892 |

Table 4: Method for contact map estimation

| Method | CP↑ | |
|---|---|---|
| | 0.05 | 0.075 |
| output meshes | 0.518 | 0.621 |
| variance estimation (ours) | 0.629 | 0.670 |

improvements. Additionally, Figure 4 depicts the geometry performance with ID and ID field volume rendering, further demonstrating the positive impact of global features.

**Ablation Study on Grouping Loss** Grouping loss function defined in Eq (12) comprises two key terms: the first is the squared distance, and the second is the exponential penalty. In Table 2, model (a) is trained without the grouping loss, while model (d) is trained with the grouping loss. The exponential function in the second term encourages soft assignment to a specific instance or cluster. However, using only this term does not lead to improved grouping performance. We observe that the combination of both terms within the grouping loss function results in overall performance enhancement.

**Contact Map Analysis** Table 1 also presents the performance of our contact map using contact precision metrics. Our approach leverages unsupervised learning to predict contact fields in 3D space, eliminating the need for labeled training data, which is often difficult and costly to obtain. This is particularly advantageous in complex scenarios involving multiple individuals and interactions. The effectiveness of our method is significantly influenced by the resolution of the 3D data. High-resolution data provide detailed and dense information, enabling precise contact prediction. In contrast, low-resolution data can lead to less accurate results due to the sparse representation of the interactions, as illustrated in Figure 5. A key advantage of our approach is the use of an implicit contact field, which allows for flexible changes in resolution. This flexibility enables our method to adapt to various data resolutions without compromising the integrity of the contact prediction. Thus, our method excels in flexibility and reduces dependency on extensively labeled datasets. However, ensuring adequate resolution of 3D data is crucial for achieving optimal accuracy and reliability in contact field estimation. In this paper, we use a resolution of $256^3$ to estimate the contact field.

To further assess our contact predictions, we directly infer contacts from the output instance meshes by examining geometric proximity and the surface identifiers used to generate the contact map from the pseudo ground truth instance meshes. Table 4 presents the results for both contact map estimation methods. Our variance-based estimation of the contact field in 3D space yields better results than the mesh-based inference. This is because the variance estimation method benefits from high-resolution 3D data and operates in continuous space, allowing for precise localization of contact areas without intermediary steps that could introduce errors. In contrast, the mesh-based inference relies on reconstructed meshes that may lack fine details due to resolution limitations or reconstruction errors, leading to less accurate contact predictions. Additional details are provided in Appendix B.3.

**Ablation Study on the Number of Views** Table 3 presents an ablation study on the number of views. Our method consistently outperforms DMC across all metrics—Chamfer Distance (CD), Point-to-Surface Distance (P2S), and Normal Consistency (NC)—in both the 4-view and 8-view settings.

## 5 Conclusion

This paper addresses the intricate challenges associated with the 3D reconstruction of multiple interacting human bodies in close proximity, an area critical for applications in virtual reality, augmented reality, robotics, and surveillance. Our approach overcomes the limitations of traditional methods that rely on models like the Skinned Multi-Person Linear (SMPL), which often struggle in scenarios with dynamic interactions and occlusions. By employing advanced methodologies, including a multi-view feature transformer and a global scene feature extraction transformer, our approach not only preserves the unique identities and spatial information of each individual but also enhances the accuracy of 3D reconstructions.

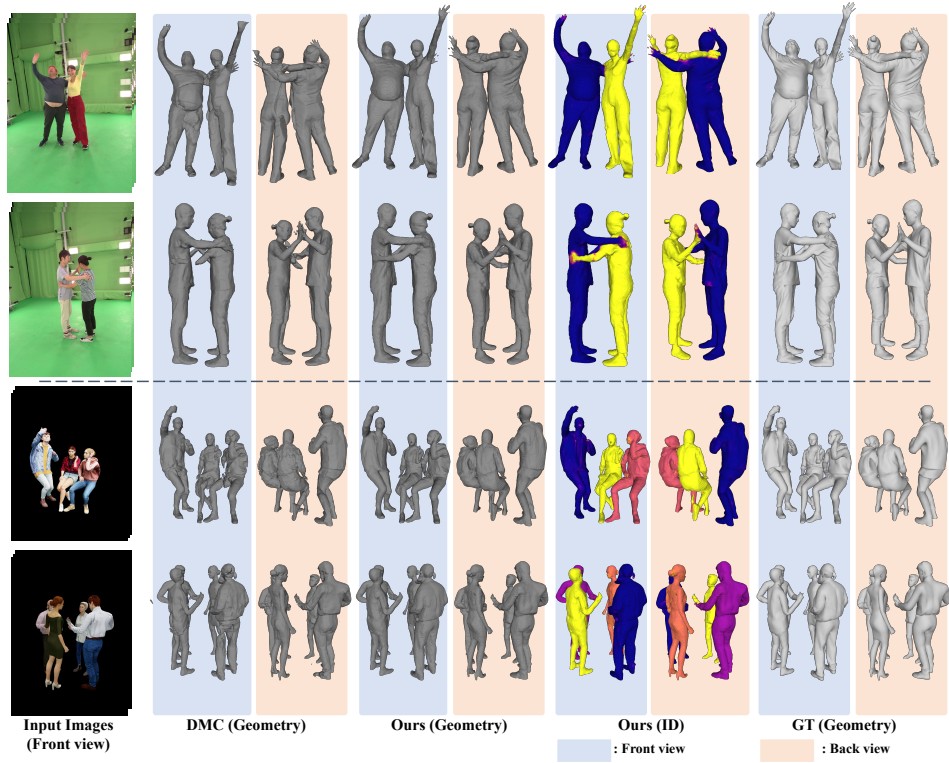

Figure 3: We compare our method to baseline DMC [43] on Hi4D (top) and SynMPI (bottom) test set. From left to right columns, we show the input multi-view images, the generated geometry by each method, and ground truth (GT) Geometry.

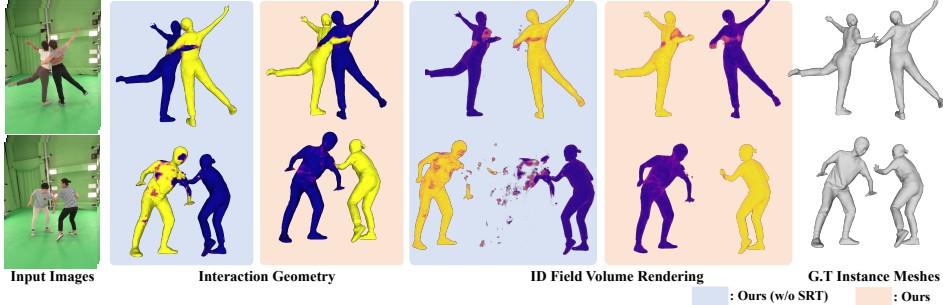

Figure 4: Visualization of reconstructed multi-person interaction geometry and instance-wise volume rendering with ID fields for visualizing occluded regions during interaction.

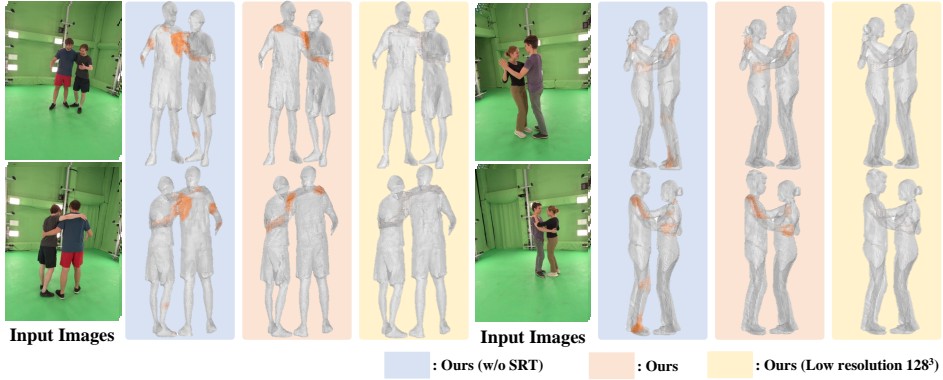

Figure 5: Comparison of the effect of global features and 3D resolution on estimated contact fields. Ours (w/o global) excludes global features, Ours (low resolution $128^3$) uses low resolution.

## Acknowledgments and Disclosure of Funding

This work was supported by Korea Creative Content Agency (KOCCA) grant funded by the Korea government (MCST) (No. RS-2023-00224427, Development of intelligent virtual sports service technology for breaking dance).

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

# A  Implementation Details

## A.1  Training

We trained our model on the Hi4D [42] dataset and our dataset SynMPI. Following the PIFu [32] process to extract pixel-aligned features, we sampled 6000 points used during training. The training was conducted with a batch size of 4 and a learning rate of $1e - 4$. Our learning rate schedule involved decaying the initial learning rate by a specified factor (gamma) at predetermined epochs, as defined in our schedule. The model was optimized using the RMSprop optimizer. In our experiments, we set $\omega_s = 1$, $\omega_{\text{contra}} = 0.1$, and $\omega_{\text{group}} = 0.1$ for the weights of loss functions. Implemented using PyTorch [29], the entire training process spanned approximately two days and covered 100 epochs using two NVIDIA A100 GPUs. Inference of each instance requires around 60 seconds on the same GPU.

## A.2  Architecture

**Multi-View Feature Fusion Module $f^{\text{e}}$.** Our objective with $f^{\text{e}}$ is to effectively aggregate features from multiple view inputs. Utilizing a view-to-view transformer architecture inspired by Double-Field [38], we process pixel-aligned features to facilitate this aggregation. The operation of our transformer is mathematically represented as:

$$
\begin{aligned}
Q_{\mathbf{x}}^f, K_{\mathbf{x}}^f, V_{\mathbf{x}}^f &= \varepsilon_{K,Q,V}^f(F_{\mathbf{x}}^1, F_{\mathbf{x}}^2, ..., F_{\mathbf{x}}^{V,}), \\
\Phi_{\mathbf{x}}^L &= \varepsilon^f(\text{Attention}(Q_{\mathbf{x}}^f, K_{\mathbf{x}}^f, V_{\mathbf{x}}^f)),
\end{aligned}
\tag{15}
$$

where $Q_{\mathbf{x}}^f$, $K_{\mathbf{x}}^f$, and $V_{\mathbf{x}}^f$ denote the query, key, and value matrices generated from the input features, respectively. Following self-attention, the features are further refined through feed-forward networks $f^F$ to obtain the local feature set $\Phi_{\mathbf{x}}^L$.

**SRT Encoder $f^{\text{se}}$.** SRT Encoder, denoted as $f^{se}$, aims to encapsulate 3D scene information into a comprehensive set-latent scene representation $z$. Following methodologies similar to those described by Object Scene Representation Transformer (OSRT) [34], our encoder leverages the Transformer's self-attention mechanism to aggregate spatial and feature information from multiple views into a single scene representation. This process is formalized as:

$$
\{F_v\}_{v=1}^V = \varepsilon_f(\{F^v, \varepsilon_{\text{ray}}(\mathbf{o}^v, \mathbf{d}^v)\}_{v=1}^V),
\tag{16}
$$

where $\varepsilon_f$ and $\varepsilon_{\text{ray}}$ represents the conv block yielding a set of features $\{F_v\}_{v=1}^V$. Subsequently, these features are aggregated into a set of flatted patch embeddings $\{E_i\}_{i=1}^N$, where $N$ denotes the total number of patches across all images. This aggregation can be mathematically represented as:

$$
\{E_i\}_{i=1}^N = \varepsilon_{\text{patch}}(\{F_v\}_{v=1}^V),
\tag{17}
$$

The transformer encoder, $\mathcal{T}^e$, then processes this set of embeddings to generate the final set-latent scene representation $z$:

$$
z = \mathcal{T}^e(\{E_i\}_{i=1}^N),
\tag{18}
$$

where $z$ fully encapsulates the observed 3D scene, encoding comprehensive spatial and visual scene information. It is imperative to note that set-latent scene representation $z$ embodies the comprehensive understanding of the specific 3D scene as observed through the corresponding set of images. This representation, characterized by its ability to maintain the integrity and richness of scene's spatial and feature information, is pivotal for subsequent reconstruction and analysis tasks.

**Global Feature Decoder $f^{\text{sd}}$.** We employ SRT decoder to extract global features from scene representation $z$. There are some differences with original SRT decoder [36]. The main difference is that local feature $\Phi_{\mathbf{x}}^L$ is used for query and value in multi-head attention mechanism. This modification ensures that:

$$
\Phi_x^G = f^{sd}(z, \Phi_{\mathbf{x}}^L),
\tag{19}
$$

where $\Phi_x^G$ represents the globally decoded feature. This enables the decoder to dynamically focus on relevant scene information, thereby facilitating detailed and accurate 3D reconstructions and precise occupancy and identification predictions.

## A.3 Dataset Construction

To create our dataset, we first acquired characters of various ages and interaction motion sequences from Character Creator 4 [31]. We then composed scenes featuring multiple characters using Omniverse USD Composer [27]. To facilitate the dataset generation process, we modified the Kaolin rendering tool [26] to include tasks such as multi-person normalization, enabling us to achieve the desired outputs. This process allows us to generate multi-view rendered images, mask images, instance masks, and 3D geometry.

## A.4 Evaluation Metrics

The definition of evaluation metrics are shown in below. $\mathcal{P}$ and $\mathcal{Q}$ refer to the set of 3D points.

**Chamfer Distance (CD)** This metric calculates the bidirectional disparity between points on the predicted and corresponding ground-truth mesh. It computes the euclidean distance from each point to nearest surface on other mesh. Lower value of CD metric indicates a higher fidelity of reconstruction.

$$\text{CD}(\mathcal{P}, \mathcal{Q}) = \frac{1}{|\mathcal{P}|} \sum_{\mathbf{p} \in \mathcal{P}} \min_{\mathbf{q} \in \mathcal{Q}} \|\mathbf{p} - \mathbf{q}\|^2 + \frac{1}{|\mathcal{Q}|} \sum_{\mathbf{q} \in \mathcal{Q}} \min_{\mathbf{p} \in \mathcal{P}} \|\mathbf{q} - \mathbf{p}\|^2 \tag{20}$$

**Point to Surface (P2S)** This metric computes the unidirectional distance from each point of ground-truth mesh to the nearest surface on the corresponding predicted mesh based on the closest-set euclidean distances. Lower value of P2S metric means superior reconstruction accuracy.

$$\text{P2S}(\mathcal{P}, \mathcal{Q}) = \frac{1}{|\mathcal{P}|} \sum_{\mathbf{p} \in \mathcal{P}} \min_{\mathbf{q} \in \mathcal{Q}} \|\mathbf{p} - \mathbf{q}\|^2 \tag{21}$$

**Normal Consistency (NC)** This metric computes the difference of normal vector between points on the predicted and corresponding ground-truth mesh with the nearest-set euclidean distances. It uses the bidirectional way for calculating the difference. Lower value of NC metric indicates a higher fidelity of reconstruction.

$$\text{NC}(\mathcal{P}, \mathcal{Q}) = \frac{1}{2|\mathcal{P}|} \sum_{\mathbf{p} \in \mathcal{P}} \left(1 - \mathbf{n_p} \cdot \mathbf{n}_{\text{nearest}(\mathbf{p}, \mathcal{Q})}\right) + \frac{1}{2|\mathcal{Q}|} \sum_{\mathbf{q} \in \mathcal{Q}} \left(1 - \mathbf{n_q} \cdot \mathbf{n}_{\text{nearest}(\mathbf{q}, \mathcal{P})}\right) \tag{22}$$

**Contact Precision (CP)** We first identify sample points in the voxel grid using the ground truth mesh $\mathcal{M}_{\text{gt}}$,

$$\mathbf{P}_{\text{inside}} = \{\mathbf{p} \in \mathbf{P} \mid \mathcal{M}_{\text{gt}} \text{ contains } \mathbf{p}\}. \tag{23}$$

For each point $\mathbf{p}_i \in \mathbf{P}_{\text{inside}}$, we determine the nearest points on the mesh surface $\mathcal{M}$ and assign surface identifiers $s_i$ based on face indices $f_i$. Using a KD-Tree, we find neighboring points within a specified distance threshold $\epsilon$,

$$N_i = \{\mathbf{p}_j \mid \|\mathbf{p}_i - \mathbf{p}_j\| < \epsilon \text{ and } j \neq i\}. \tag{24}$$

A contact is marked if any neighboring point has a different surface identifier,

$$\text{contact}_i = \begin{cases} 1 & \text{if } \exists \mathbf{p}_j \in N_i \text{ such that } s_j \neq s_i \\ 0 & \text{otherwise} \end{cases}. \tag{25}$$

This process results in contact labels that are contextually relevant to the mesh's surface features, enabling validation of our contact prediction algorithms using precision as the evaluation metric.

The contact precision is defined by the overlap between the estimated contact map $\mathbf{E}$ and the pseudo ground truth contact map $\mathbf{T}$ generated from ground truth meshes. Then, the precision is given by

$$P(\mathbf{T}, \mathbf{E}) = \frac{|\mathbf{T} \cap \mathbf{E}|}{|\mathbf{E}|}, \tag{26}$$

where $|\mathbf{T} \cap \mathbf{E}|$ is the number of true positives (correctly predicted contact points), and $|\mathbf{E}|$ is the total number of predicted contact points. A higher precision value indicates that a greater proportion of the contact points predicted by the model are correct, signifying fewer false positives. This metric is crucial for assessing the accuracy of our contact prediction algorithms, ensuring that the predicted contacts closely align with the contacts defined by the pseudo ground truth.

Table A: Evaluation excluding SynMPI dataset on Hi4D test sets

| | Geometry | | | Contact Precision ↑ | | | |
|---|---|---|---|---|---|---|---|
| | CD↓ | P2S↓ | NC↑ | 0.025 | 0.05 | 0.75 | 0.1 |
| w/o synthetic data | 0.499 | 0.418 | 0.885 | 0.351 | 0.482 | 0.514 | 0.542 |
| w/ synthetic data | **0.406** | **0.329** | **0.892** | **0.447** | **0.629** | **0.670** | **0.703** |

Table B: Ablation study on SPML initialization method for synthetic datasets in DMC

| Model | SMPL method | CD↓ | P2S↓ | NC↑ |
|---|---|---|---|---|
| DMC | MVPose [11] | 0.805 | 0.489 | 0.771 |
| DMC | LVD [10] | 0.631 | 0.495 | 0.768 |

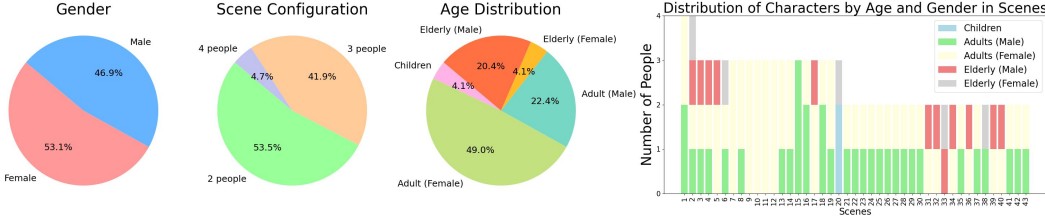

Figure A: Statistics of the SynMPI dataset.

# B Analysis

## B.1 Datasets

Table A presents the results of an ablation study using our SynMPI dataset. Training models with SynMPI alongside Hi4D [42] led to improved performance on the Hi4D test set. This improvement is mainly due to the large diversity within our dataset, which includes variations in age, gender, and scene composition, as shown in Figure A.

## B.2 Baseline Models

For the baseline models discussed in Section 4.3, we employed different SMPL [18] acquisition methods tailored to each dataset. Specifically, for the HI4D dataset, we used MVPose [11], following the experimental setup described in the HI4D paper, where DMC [43] was run using MVPose for SMPL acquisition. We adopted this approach to ensure consistency and comparability.

Table B presents the ablation study on initial SMPL methods. However, we encountered challenges when using MVPose for our synthetic dataset, as its modules were trained on real data, leading to less accurate SMPL estimations on synthetic data. To address this, we utilized Learned Vertex Descent (LVD)[10], which is designed to fit SMPL to a 3D human model (3D scan) and has proven to provide more accurate results in this context. Although LVD is originally intended for single-person scenarios and may be sensitive to occlusions, we selected it for the synthetic dataset to achieve accurate results in our study.

## B.3 Contact map

We also measure contact precision using the generated instance meshes as well as our proposed variance estimation of contact fields. To provide further insights into our method, we explain how different individuals are distinguished using predicted ID values during mesh generation.

To generate instance meshes from our implicit fields, our algorithm identifies and marks regions of interest based on the occupancy field during inference, excluding the background. This process generates both an ID field and a contact field. The normalized ID values within these regions

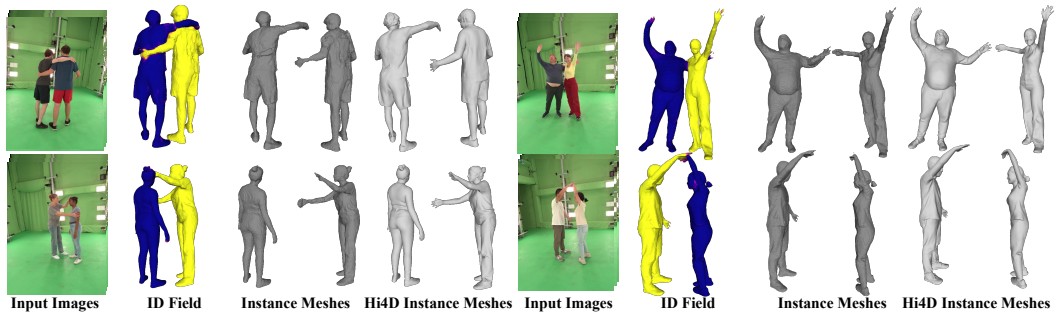

Figure B: Example of Instance meshes.

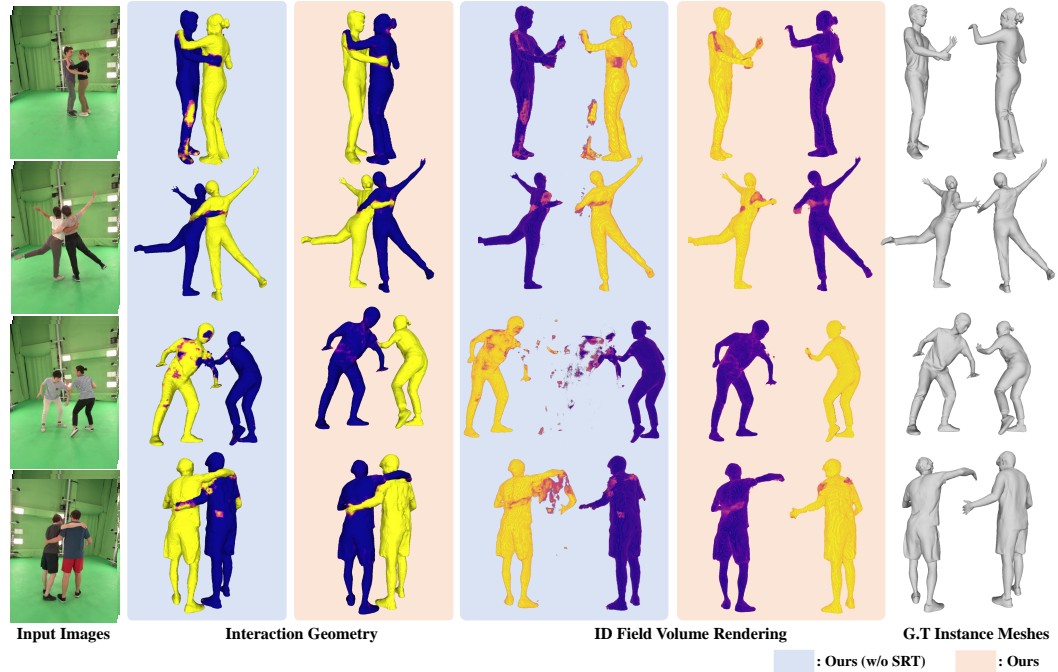

Figure C: Visualization of reconstructed multi-person interaction geometry and instance-wise volume rendering with ID fields for visualizing occluded regions during interaction.

are processed using $k$-Means clustering [17], grouping the data into clusters, with each cluster representing a different individual and including the associated contact regions.

After clustering, each cluster is isolated with a binary mask, which is smoothed using a Gaussian filter to create a blending mask that ensures smooth transitions at boundaries. This blending mask is applied to the occupancy field to enhance boundary details. Finally, the marching cubes algorithm generates a 3D mesh for each cluster from the processed occupancy field. These steps allow us to reconstruct instance meshes that support further evaluation of contact predictions.

Figure B illustrates the results of our instance meshes.

## C   Results

### C.1   Visualization

We present additional visualization result samples in Figures D, E, G, and H. Additionally, Figure C shows further visualization results from our ablation study on architecture. Figure F provides more examples comparing contact maps based on 3D space resolution.

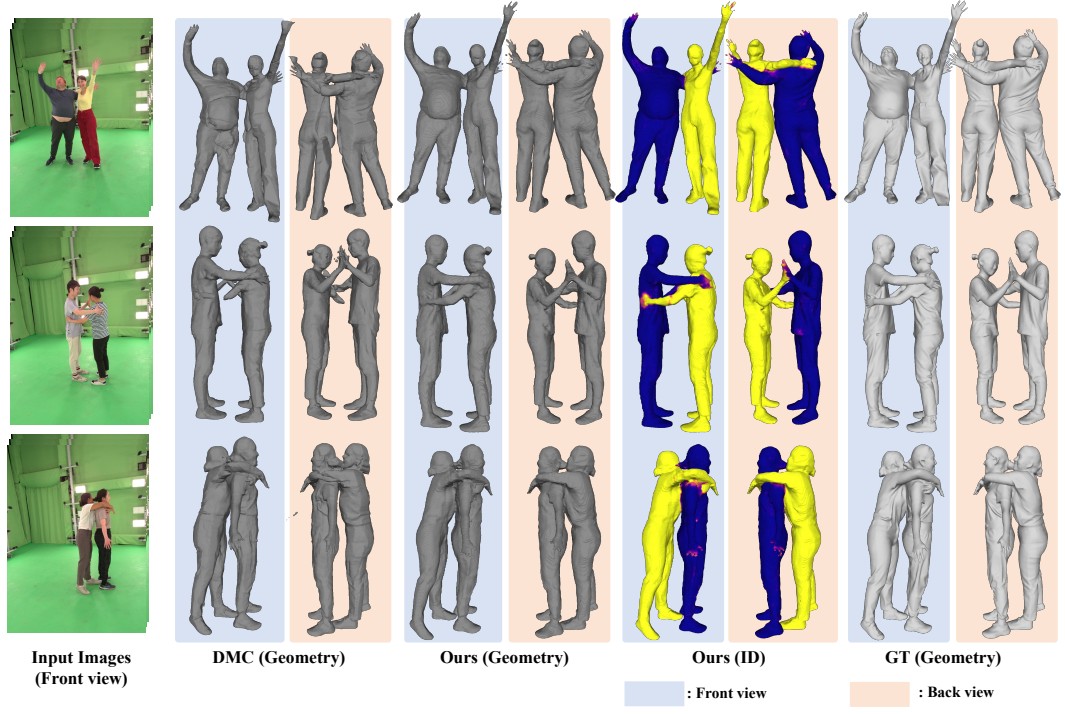

**Input Images (Front view)** · **DMC (Geometry)** · **Ours (Geometry)** · **Ours (ID)** · **GT (Geometry)**

: Front view · : Back view

Figure D: Extended visualization of our method compared to baseline DMC [43] on Hi4D test split.

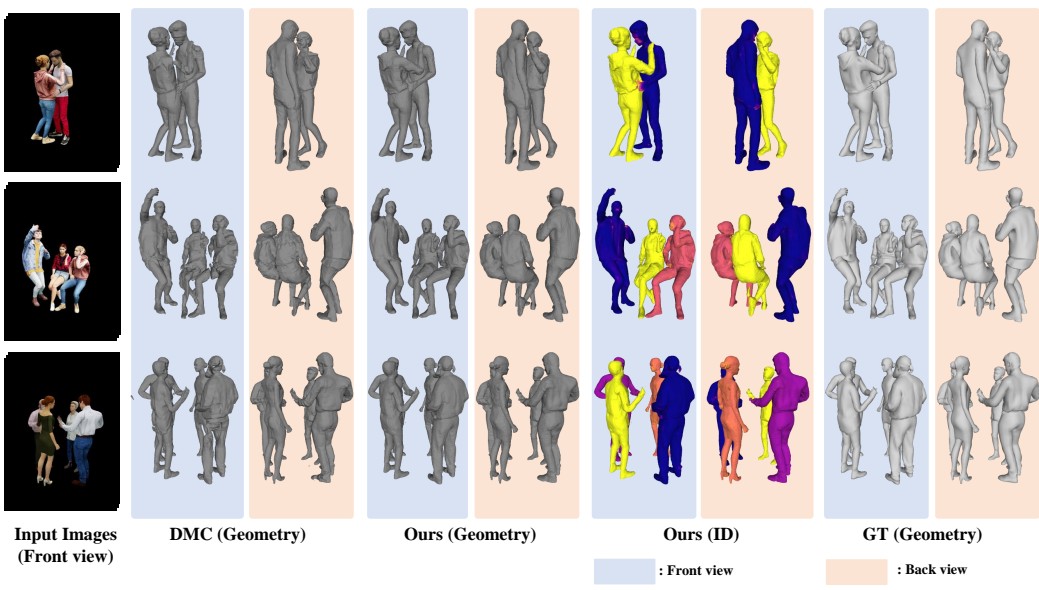

**Input Images (Front view)** · **DMC (Geometry)** · **Ours (Geometry)** · **Ours (ID)** · **GT (Geometry)**

: Front view · : Back view

Figure E: Extended visualization of our method compared to baseline DMC [43] on test split of our synthetic dataset.

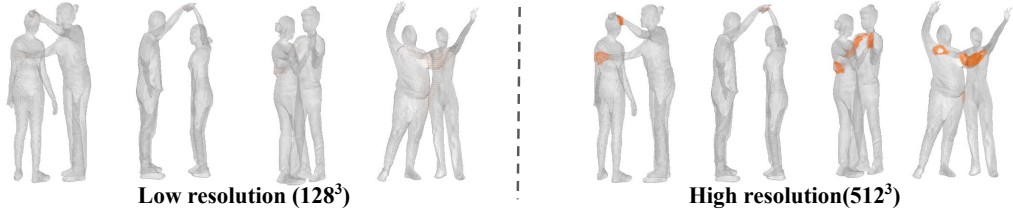

**Low resolution (128³)**      **High resolution(512³)**

Figure F: Example of low resolution contact maps.

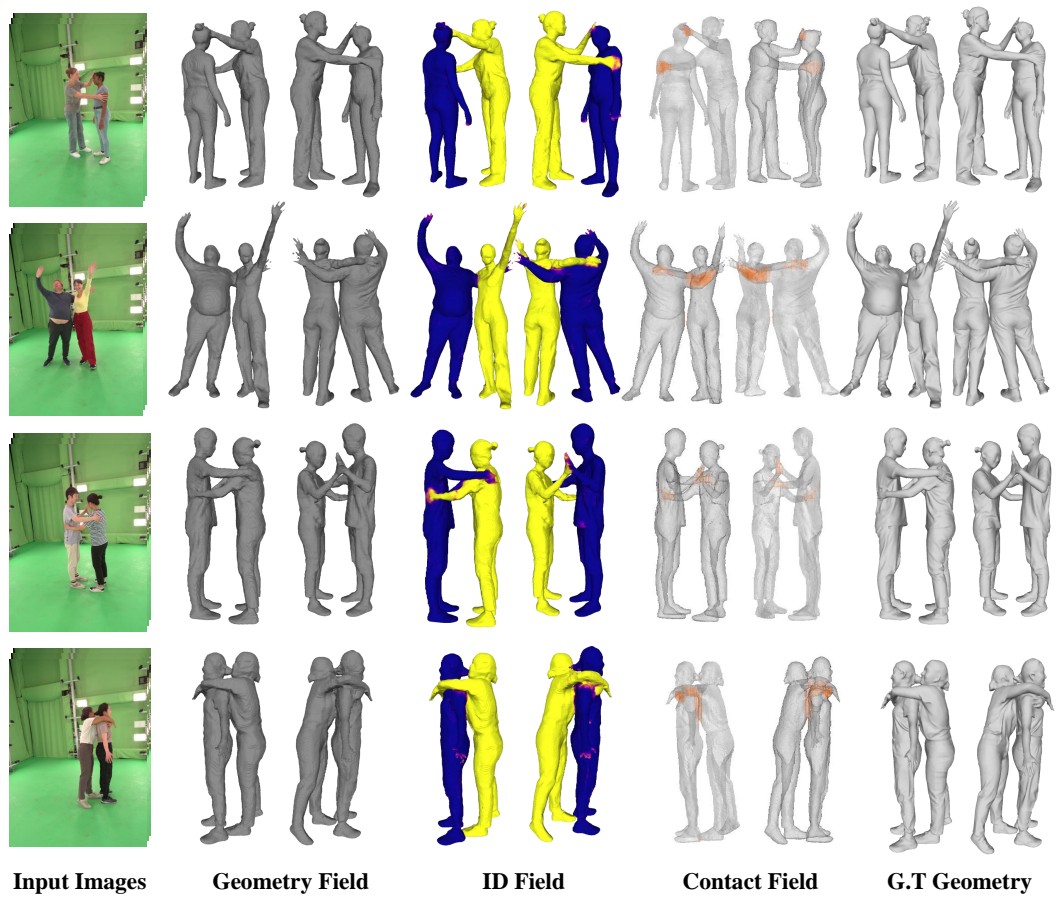

**Input Images**     **Geometry Field**     **ID Field**     **Contact Field**     **G.T Geometry**

Figure G: Contact Field results of ours in Figure. 3

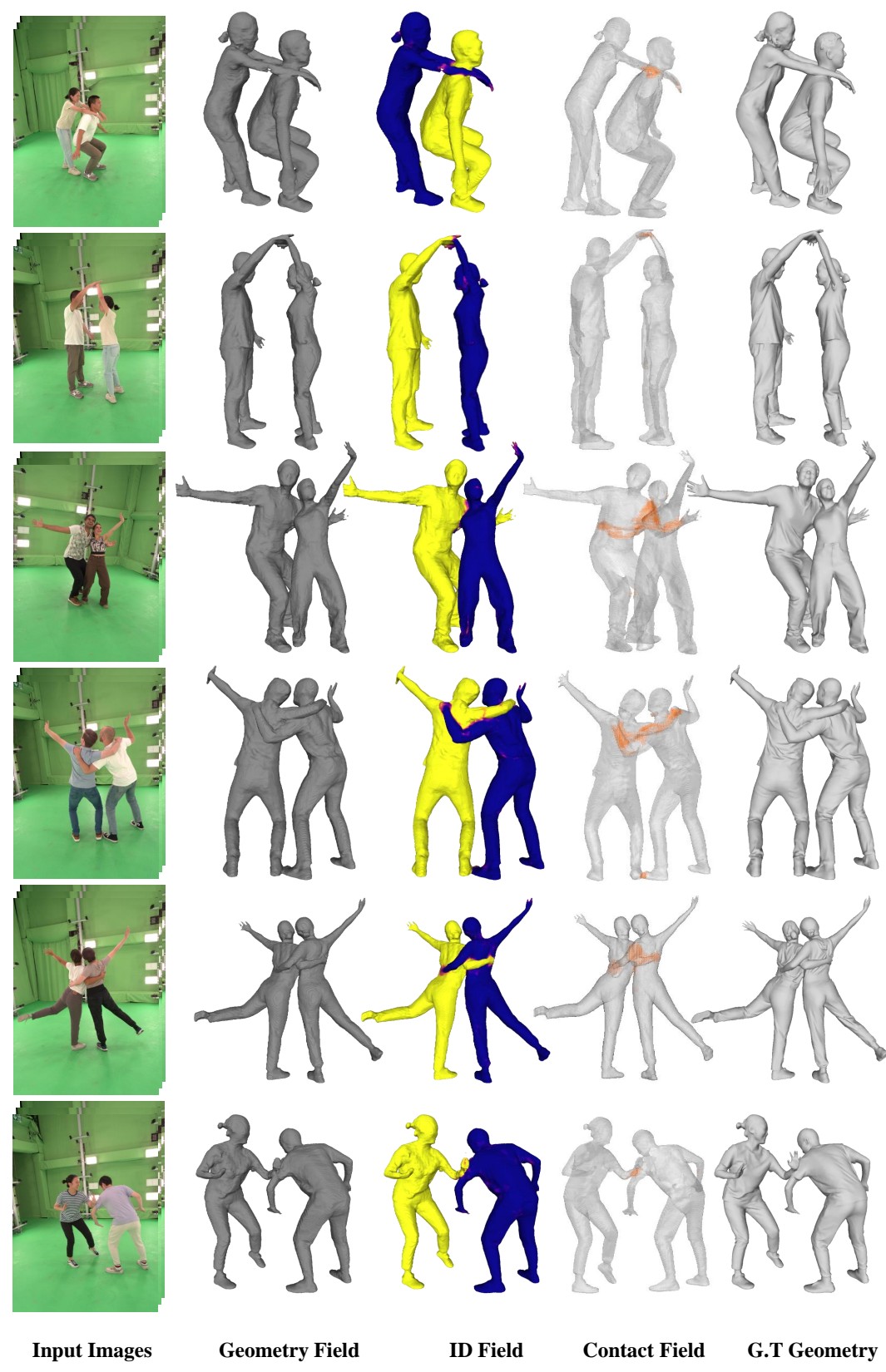

| Input Images | Geometry Field | ID Field | Contact Field | G.T Geometry |
|---|---|---|---|---|

Figure H: Additional results of our method ContactField in Hi4D test split.

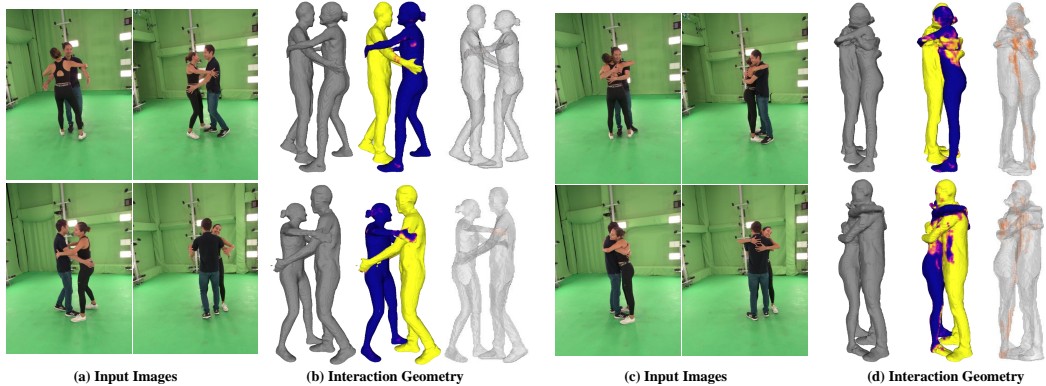

| (a) Input Images | (b) Interaction Geometry | (c) Input Images | (d) Interaction Geometry |

Figure I: Failure case.

# D    Discussion

## D.1    Limitation

Despite its effectiveness, our approach introduces certain limitations, such as resolution constraints when reconstructing all elements simultaneously, which can affect the finer details of the models. Especially, we present failure cases in (d) of Figure I. Since we do not incorporate spatial prior such as SMPL, predicting identity (ID) in extreme poses, such as hugging, becomes challenging. However, as shown in (b) from a few frames earlier, the prediction is accurate. This suggests that incorporating a temporal module could be a promising direction for future work. Still, experimental results affirm the superiority of our approach over traditional methods, indicating significant potential for future enhancements in complex interaction scenarios and larger group dynamics. Moving forward, we aim to refine our techniques to address these resolution limitations and explore broader applications, further advancing the realism and functionality of 3D human body reconstructions.

# E    Broader Impacts

This novel implicit field representation for multi-person interaction geometry in 3D spaces has the potential to advance various applications in healthcare, sports, and security by enabling more accurate and detailed reconstructions of human interactions. However, the enhanced ability to capture such interactions also raises important concerns regarding privacy and ethical use. To mitigate these risks, it is essential to implement robust data protection measures, establish clear ethical guidelines, and ensure compliance with privacy laws.

