# OpenReview forum: "ContactField: Implicit Field Representation for Multi-Person Interaction Geometry"
_NeurIPS.cc/2024/Conference — NeurIPS 2024 poster_

### Official Review · Reviewer_N9qi · 2024-06-21

**Soundness:** 3
**Presentation:** 3
**Contribution:** 3
**Rating:** 5
**Confidence:** 3

**Summary:**

A novel implicit field representation is designed for multi-person geometry modeling, which manages to estimate the occupancy, identity, and geometry simultaneously. Moreover, to alleviate the occlusion issue, an additional 3D scene representation module is designed. A synthetic dataset with multi-view multi-human interaction is developed. Experiments show the superiority of the proposed method.

**Strengths:**

- Estimating the contact fields by the variance of the predicted ID field is an interesting idea and has shown effectiveness.

- The performance is impressive both quantitatively and qualitatively.

**Weaknesses:**

- More details on the proposed synthetic dataset should be provided, including the data source, the synthesis and annotation protocol, and the data scale.

- Some interested hyper-parameters are not specified, including the contact deviation threshold \tau_c in L236.

**Questions:**

- The aspect ratio for the data sample videos seems not right.

- In inference, how are the different persons separated by the predicted ID values in detail? Is the number of existing human instances required as input?

- How is the local neighborhood \mathcal{N}(v) decided in L233?

**Limitations:**

The limitations have been discussed by the authors.

---

> ### Author Rebuttal · Authors · 2024-08-07
>
> Thank you for your valuable feedback on the synthetic dataset and our experiments. Below are our responses to weaknesses and questions in your comments.
>
> ## **Weakness 1:** details about datasets
> We outline the relevant information about the dataset below.
>
> **Data Source**:
> Initially, we acquired characters with various ages and interaction motion sequences from Character Creator 4 [1].
>
> **Synthesis and Annotation Protocol**:
> Subsequently, we composed scenes featuring multiple characters using Omniverse USD Composer [2]. To facilitate the dataset generation process, we engineered the Kaolin rendering tool[2], incorporating tasks such as multi-person normalization to achieve the desired outputs. This process enabled us to generate multi-view rendered images, mask images, instance masks, and 3D geometry. Additionally, we created normal ground truth maps and joints, although these were not utilized in this paper.
>
> **Data Scale**:
> The dataset includes 49 characters (26 female, 23 male) with 50 motion sequences across 43 scene configurations, totaling approximately 25,000 frames. For a detailed breakdown, please refer to Figure A in the attached PDF file.
> We hope this detailed information will provide a clearer understanding of our synthetic dataset and its role in evaluating our proposed method.
>
> ## **Weakness 2:** detailed hyper-parameters
> The hyper-parameters are as follows: $\tau_{c}=0.25$ in L236, $\omega_{s} = 1, \omega_{\text{contra}} = 0.1 \omega_{\text{group}}=0.1$ in equation (9). We will ensure these details are included in the revised version of the paper. Thank you for your careful review and pointing out the missing hyper-parameters.
>
> ## **Question 1:** aspect ratio
> Thank you for your observation regarding the aspect ratio of the data sample videos.   The aspect ratio of figures in the paper is aligned with benchmark datasets such as Hi4D. However, it appears that the sample video had the contact field rendered with a different aspect ratio compared to the other results. We have re-adjusted the aspect ratio and created a new video. Refer to adjusted frame examples in Figure E. of the attached PDF.
>
>
> ##  **Question 2**
> Here is a detailed explanation of how different people are distinguished using predicted ID values:
>
> **1. Region Identification**: During inference, the algorithm identifies and marks regions of interest based on the occupancy field, excluding background regions. This step generates both an ID field and a contact field.
>
> **2. Clustering**: The normalized ID values within these regions are processed using a clustering algorithm, specifically K-Means. This algorithm groups the data into unique clusters, each representing a different person, and includes the contact regions for all individuals.
>
> **3. Segmentation**: After clustering, each cluster is isolated with a binary mask. A binary mask is then created and smoothed with a Gaussian filter to form a blending mask, ensuring gradual transitions at boundaries. This blending mask is applied to the occupancy field to enhance boundary details. Finally, the marching cubes algorithm generates a 3D mesh model for each cluster from the processed occupancy field.
>
> Instance mesh visualization results are presented in Figure B. of the attached pdf file.
>
> ## **Question 3:** local neighborhood
> The local neighborhood $N(v)$ is defined as the set of points within a specified radius (contact threshold) around the point $v$, excluding $v$ itself, and is determined using a KDTree [4] for efficient querying. The criteria for marking a point as a contact is based on the presence of neighboring points with differing surface identifiers.
>
> If you have any concerns or questions, feel free to leave comments.
>
> References
>
> - [1] Character Creator 4, https://www.reallusion.com/character-creator/
>
> - [2] NVIDIA Omniverse USD Composer, https://docs.omniverse.nvidia.com/composer/latest/index.html
>
> - [3] NVIDIA Kaolin, https://github.com/NVIDIAGameWorks/kaolin
>
> - [4] Bentley, Jon Louis. "Multidimensional binary search trees used for associative searching." Communications of the ACM. 1975.

---

> > ### Comment · Reviewer_N9qi · 2024-08-12
> >
> > Thanks for the responses. All my concerns are addressed.

---

> > > ### Author Response · Authors · 2024-08-12
> > >
> > > I am glad to hear that your concerns have been resolved. Thank you once again for your insightful feedback on our work. Your comments have been invaluable in improving our paper. If you have any further questions or additional comments, please do not hesitate to reach out.

---

### Official Review · Reviewer_Vhig · 2024-07-04

**Soundness:** 2
**Presentation:** 3
**Contribution:** 2
**Rating:** 4
**Confidence:** 4

**Summary:**

This paper proposes a method to reconstruct close interactions from multi-view images using an implicit representation. The occupancy and ID fields are directly regressed from multi-view images, which can then be used to infer contacts. To fuse multi-view information, a transformer-based module is introduced to consider both local and global features. The authors also built a synthetic dataset to assist in training. The experimental results show that the performance is good.

**Strengths:**

- The proposed dataset may be useful for future tasks involving close interaction reconstruction.
- The qualitative results are good.

**Weaknesses:**

- The underlying idea is almost the same as DeepMultiCap [39], which also represents interactive humans with pixel-aligned implicit functions and adopts a transformer to fuse multi-view information. Compared to DeepMultiCap, the proposed method does not leverage prior human knowledge and may induce artifacts in occluded body parts. For occluded and interactive body parts, the proposed method cannot sample valid pixel-aligned features. I am confused about the proposed method's ability to address heavy occlusion without any prior human knowledge.
- The framework is completely implicit like PIFU; however, its generalization ability may not be good due to the high-dimensional output.
- It seems contacts can also be inferred from the output meshes as pseudo ground-truth contact generation. What is the difference between this approach and the proposed variance estimation?

**Questions:**

- How can a different number of input views affect reconstruction performance?
- What is the generalization ability of the proposed method on novel subjects and different camera extrinsics?
- The ability to address occlusions should be clarified.
- Why not use contact information provided in Hi4D for contact evaluation?

**Limitations:**

The limitations and societal impact are discussed in the paper.

---

> ### Author Rebuttal · Authors · 2024-08-07
>
> Thank you for your valuable question. Below are our responses to raised weaknesses and questions.
> ## **Weakness 1:** multi-person interaction with SMPL
> Thank you for your valuable feedback. In the case of DeepMultiCap (DMC) [1], which uses the SMPL human prior, reconstruction is performed one person at a time. When body parts are occluded, they are masked and included in the input. Using SMPL is particularly helpful in handling occlusions when many body parts are not visible, as it follows the human shape. For areas where body parts are not visible, DMC relies more on the SMPL shape itself than on pixel-aligned features. As the below table shows, the reconstruction performance varies for results from each SMPL method.
>
> | Model setting         | CD↓| P2S↓ | NC↑|
> |-----------------------|---|----|----|
> | DMC  (w. syn(MVPose)) | 0.805 | 0.489 | 0.771 |
> | DMC  (w. syn(LVD)) | 0.631 | 0.495 | 0.768 |
>
> In extreme poses or close interaction scenarios, estimating SMPL becomes challenging and requires an optimizing process for accurate estimation, which is time-consuming. Additionally, SMPL has limitations in broadly representing the human shape, especially when children are present in our synthetic datasets. SMPL struggles to accurately represent children, making training difficult.
> We also considered expanding to interactions with entire scene objects, which raised concerns about using SMPL. Therefore, we aimed to handle occlusions by reconstructing the entire geometry from multi-view inputs and predicting the ID field in 3D space using SRT features. By predicting contact based on ID, our method aims to understand multiple people in close interaction scenarios. Additionally, we created and trained our model on synthetic data with numerous instances of self-occlusion and multi-person interaction using benchmark datasets to ensure robust SRT feature extraction.
>
> - [1] Yang Zheng, Ruizhi Shao, Yuxiang Zhang, Tao Yu, Zerong Zheng, Qionghai Dai, and Yebin Liu. DeepMultiCap: Performance capture of multiple characters using sparse multiview cameras. In Proceedings of the IEEE/CVF International Conference on Computer Vision. 2021.
>
>
> ## **Weakness 3:** contact information from the output meshes
> Thank you for your insightful comments regarding the generation of pseudo ground-truth contacts from output meshes and the proposed variance estimation method for contact detection. Contact label generation relies on geometric proximity and surface identifiers to infer contacts directly from the mesh, while Estimated Contact Fields use statistical variance to detect contacts within the 3D space.
> To address your comment, we have inferred contacts from the output meshes using the contact label generation approach and measured the precision of our contact predictions. Examples of instance meshes are presented in Figure B in the attached PDF file.
>
> |         | CP $( \\delta = 0.05 $)↑ | CP ($ \\delta = 0.075 $)↑ |
> |-----------|---|---|
> | output meshes (requested) | 0.518 | 0.621 |
> | variation estimation (ours) | 0.629 | 0.670 |
>
>
> ## **Question 1:** the performance of different number of input views
> We will report the performance of our model with smaller views before the end of discussion period.
>
> ## **Weakness 2 and Question 2:** generalization ability on novel object or different extrinsic camera parameter
> To thoroughly evaluate this aspect, we conducted two experiments using our pre-trained model. These experiments included zoom-in and zoom-out tests with the Hi4D dataset, as well as tests on synthetic data featuring four individuals performing extreme poses, such as breakdancing, introducing novel postures and configurations unseen during training with new camera settings. Qualitative results are provided in Figures C and D of the attached PDF.
> During the zoom-in and zoom-out tests on Hi4D, we adjusted the images and camera parameters accordingly. In the zoom-out scenario, the person's size in the image decreased, leading to a loss of detail in feature extraction.
> However, our implicit fields prevent significant performance degradation by taking the 3D space bounding box around the person's center, thereby preserving the geometrical results. The qualitative results, shown in Figure C. indicate that the model can handle previously unseen zoom-in and zoom-out scenarios. Notably, the model performed better during zoom-out than zoom-in for ID field prediction, as it relies on extracting features from the entire scene.
>
> ## **Question 3:** robustness on occlusion
> Our approach addresses occlusions by presenting interaction geometry. We estimate complete geometries from multiple views and use SRT features to predict IDs and contact regions in 3D space, even in occluded areas. By predicting contact regions based on IDs, our model handles occlusions in close interaction scenarios as shown in the first row of Figure B in the attached PDF file.
> Additionally, we created synthetic data with numerous multi-person interactions and trained our model using this data alongside benchmark datasets to enhance robustness in extracting SRT features.
>
> ## **Question 4:** Why not using contact information of Hi4D
> To the best of our knowledge, while the Hi4D dataset provides SMPL contact ground truth (G.T.) information, we could not find any contact G.T. information specifically for the 3D scans in the Hi4D dataset.
> As a result, we generated a pseudo ground truth using the 3D instance mesh provided by Hi4D and used it for our measurements, as detailed in the paper. We appreciate your question and will include a clarification on this point in the revised manuscript to ensure transparency.
> There may be misunderstandings in our responses, so it would be great if we could continue the discussion regarding any unclear or intriguing parts of our answers or any unresolved questions.
>
> If you have any concerns or questions, feel free to leave comments.

---

> > ### Author Response · Authors · 2024-08-12
> >
> > ### **Question 1** the performance of different number of input views
> >
> > We conducted a series of experiments on the Hi4D dataset to compare our method with DMC using 4 input views. The results are summarized below:
> >
> > | Input views | Model | CD↓ | P2S ↓| NC↑ |
> > |---|-----------------------|---|----|----|
> > | 4-view | DMC  | 1.304 | 0.922 | 0.705 |
> > | 4-view | Ours  | 0.761 | 0.472 | 0.870 |
> > | 8-view | DMC   | 0.631 | 0.495 | 0.768 |
> > | 8-view | Ours   | 0.406 | 0.329 | 0.892 |
> >
> > Our framework consistently outperforms DMC across all metrics in both 4-view and 8-view settings. Although performance decreases for both models when the number of views is reduced from 8 to 4, our method exhibits a smaller decline in accuracy and surface detail compared to DMC.
> >
> > The performance drop in Chamfer Distance (CD) with fewer views is primarily due to reduced object coverage. With only 4 views, the model receives less information for accurate shape reconstruction, which may result in less precise outcomes. This reduction in input data can lead to more scattered or inaccurate point predictions around the person, contributing to a higher CD.
> >
> > I hope this response, along with our previous rebuttal, satisfactorily addresses your concerns. If you have any unresolved questions or other concerns, please feel free to ask.

---

> > ### Comment · Reviewer_Vhig · 2024-08-13
> >
> > Thanks for the rebuttal. One remaining question is how many subject pairs were used in the training and testing, respectively? Can the trained model generalize to subjects outside the Hi4D and synthetic datasets (e.g., CHI3D and MultiHuman proposed in DMC)?

---

> > > ### Author Response · Authors · 2024-08-14
> > >
> > > **Training and Testing Set Composition**
> > >
> > > The Hi4D dataset comprises 40 subjects, organized into 20 unique pairs, resulting in a total of over 11,000 frames. These subject pairs (16 female, 24 male) exhibit a wide range of physical characteristics, including varying heights, weights, and garments.
> > > In our synthetic dataset, we created 49 distinct characters (26 female and 23 male) forming 43 unique pairs, which include groups of 2, 3, and 4 people. To enhance the diversity seen in Hi4D, we generated synthetic data that represents individuals of various ages, along with different heights, weights, and garments. Detailed statistics are provided in Figure A. The synthetic dataset contains approximately 25,000 frames.
> > >
> > > Consistent with existing human reconstruction settings using multi-view images [1,2], we randomly selected 521 static frames from Hi4D and 504 static frames from the synthetic dataset, using full frames. The test sets, separate from the training sets, included 150 frames from Hi4D and 90 frames from the synthetic dataset, excluding any subjects without contact.
> > >
> > > **Beyond Hi4D and the Synthetic Dataset**
> > >
> > > Figure D in the attached PDF visualizes entirely new scene configurations, featuring new subject configurations, extreme and novel poses captured from breakdancing motion sequences, and new camera settings, including different resolutions and extrinsic and intrinsic parameters. These settings were not used in the proposed synthetic dataset, demonstrating our model’s ability to generalize to new subjects.
> > >
> > > We appreciate your interest in generalization experiments with non-synthetic datasets. Due to time constraints, we were unable to fully process and experiment with the MultiHuman or CHI3D datasets, including the rendering needed to create input images. We plan to extend our model to cover these benchmarks in a future version of the manuscript.
> > >
> > > We understand your concern regarding generalization, especially since our model is trained from scratch solely from data without using prior knowledge, such as a human prior model from SMPL. In this regard, we believe our approach could benefit from prior knowledge and improve generalization by incorporating features from pre-trained models like DINOV2[3], which inherently include pose and depth information learned from image- and patch-level matching supervision. We will explore this in future research.
> > >
> > > Thank you once again for your valuable comments.
> > >
> > > References
> > > - [1] Shao, Ruizhi, et al. "Diffustereo: High quality human reconstruction via diffusion-based stereo using sparse cameras." European Conference on Computer Vision. Cham: Springer Nature Switzerland, 2022.
> > > - [2] Zheng, Shunyuan, et al. "Gps-gaussian: Generalizable pixel-wise 3d gaussian splatting for real-time human novel view synthesis." Proceedings of the IEEE/CVF Conference on Computer Vision and Pattern Recognition. 2024.
> > > - [3] Oquab, Maxime, et al. "Dinov2: Learning robust visual features without supervision." arXiv preprint arXiv:2304.07193 (2023).

---

> > > > ### Comment · Reviewer_Vhig · 2024-08-14
> > > >
> > > > Is there any overlap between the subjects in the train and test sets in Hi4D dataset?

---

> > > > > ### Author Response · Authors · 2024-08-14
> > > > >
> > > > > Yes, there are overlaps in subjects between the training and test sets because we randomly selected sequences from the Hi4D dataset for both sets, following the established protocols [1, 2] for reconstructing humans from multi-view images. However, there is no overlap in the motion sequences of these subjects.
> > > > >
> > > > > Due to only a single day left for experiments, we could only refer to new synthetic samples in the provided sample Figure D in attachment. Those samples are rendered from completely novel configurations of characters and motions used in the main paper. We will extend our model to cover the mentioned benchmarks (Multihuman, CHI3D) in a future version of the manuscript.
> > > > >
> > > > > Thank you for your detailed question and discussion.
> > > > >
> > > > > **References**
> > > > > - [1] Shao, Ruizhi, et al. "Diffustereo: High quality human reconstruction via diffusion-based stereo using sparse cameras." European Conference on Computer Vision. Cham: Springer Nature Switzerland, 2022.
> > > > > - [2] Zheng, Shunyuan, et al. "Gps-gaussian: Generalizable pixel-wise 3d gaussian splatting for real-time human novel view synthesis." Proceedings of the IEEE/CVF Conference on Computer Vision and Pattern Recognition. 2024.

---

### Official Review · Reviewer_Ar19 · 2024-07-06

**Soundness:** 3
**Presentation:** 3
**Contribution:** 3
**Rating:** 7
**Confidence:** 4

**Summary:**

The paper focuses on the problem of multi-person reconstruction from multi-view images in the face of close interactions (e.g., in cases where are in contact). The objective of this work is to propose a method to reconstruct one mesh per person that is able to capture both fine grained details (e.g., garments, face, hands, etc.) and the contact points between each of these meshes if in contact. The paper introduces a new implicit field representation specially designed for the multi-person setting. This representation allows the reconstruction of occupancy, instance ID (i.e., to which mesh the queried point corresponds), and contact fields at the same time. Aside from the occupancy, the variables for ID and contacts enables to determine which meshes belong to which person so that each mesh can be treated as separate instance. The method facilitates unsupervised estimation of contact points without the need for contact annotations in the training data.

To achieve the properties mentioned above, the authors propose to enrich the implicit representation by a multi-vew transformer-based feature extraction module to retrieve a mix of local and global features. This enables the modeling of close physical interactions by dense point retrieval in small areas (by exploiting this property of implicit fields to query points at variable resolutions). The work also presents a synthetic dataset containing diverse multi-person interaction scenarios to train the model.

SoTA methods reconstruct multiple people by either: (1) capturing each person’s geometry separately or (2) merging nearby 3D geometries. In this paper, the authors are able to reconstruct multiple bodies at the same time by using the proposed augmented implicit representation.

**Strengths:**

The strengths are the following:
 * The overall quality of the paper is good. The authors clearly identify SoTA limitations, explain them and propose a sound way to deal with these limitations.
 * Overall writing is good as it communicates the ideas clearly, though, authors may need to improve the writing in some parts of the paper to fix formatting issues and typos. (Suggestions are proposed in this review).
 * The method is sound and the proposed techniques used to implement it make sense.

**Weaknesses:**

### **Ablation missing**
L217: Authors state that "bifocal approach of the loss function is crucial". This statement needs to be backed in the experiments. However, this is not included in the ablation study. I suggest authors include an ablation experiment on the rebuttal, otherwise this is an unsupported claim which weakens the paper by a lot, even to the risk of rejection.

### **More information about the dataset**
I would like to know how the dataset was created given that it is a synthetic dataset. For example, which software was used to create this. Explain the procedure and add some statistics. Not enough detail is provided here.

### **SMPL close interactions representation**
L32-33. Can the authors explain why “the need to configure the entire scene on an individual basis complicates the accurate representation of close interactions”? I understand that the SMPL model represents a naked body and thus cannot be used “as is” to reconstruct fine-grained details. However, I don’t understand why having individual instances of SMPL would complicate the representation of close interactions? I think this is not necessarily true. I agree that it does not naturally include a representation for interactions, but this could be added to the SMPL model. Now saying that it complicates the representation may be too much?

### **Baseline models**
Baseline models: For Baseline models (Sec 4.3). Could you clarify which model MVpose/LVD you use for each dataset? Do you use both for both datasets? LVD only for the synthetic one? Is one of these used by DMC? If not, why not use the one proposed by them? Also why use LVD, is this model not sensitive to occlusion and originally used for only a single person?

### **Minor details**
* L143: q E R --> q E Z+, it would be more precise as q should not take decimal values, right? It is in a sense a segmentation task. In the case it is R, then is there a threshold to classify each instance?
* L150: Are the camera parameters K^v estimated or assumed given?

### Observations related to writing
* Citations typically include the names of all authors. In this case, some citations have all the names, but for most of them, it read "et. al.". Also publication conference names are not complete in some cases. For example, L367-368, the conference reads "IEEE" which is incomplete. It should say at least "CVPR".
* L27-28. This phrase is badly formulated. First, occlusion does not "obscure" a subject. Specifically, saying "obscures subjects from view" does not make sense. I would remove that phrase from the last part of this sentence for something like "...mainly due to occlusion, which complicates the accurate…"
*  L32-33. This phrase is not very clear. I understand what it says, but it took me a second read to clarify. Specially, the part that states “configure the entire scene on an individual basis complicates the accurate representation of close interactions” is a bit confusing to me.
* L65: The name of the module "transformer-based multi-view local-global" is a mouthful. I would give a simpler name to this.
* L81: Related works, when talking about SMPL, I would remove that it does not capture facial expressions as SMPL-X does this.
* Suggestion, don't use the work necessitates, use needs instead. it is easier to read. Try to avoid fancy wording whenever possible.
* What is deficit information? Is this a formal term? If not please rephrase this “deficitary information” or something similar.
### Typos:
* L102: you could say generative model instead of "latent vectors". This gives more clarity.
* L138: typo. "using given", remove "using"
* L150: extracting pixel-aligned features followed by PIFu --> …following PIFu
* L184: use --> is used
* L223: Initially, define --> Initially, we define

**Questions:**

Please, address the questions posed in the **Weaknesses** section.

**Limitations:**

Yes, the authors include limitations in the supplementary material.

---

> ### Author Rebuttal · Authors · 2024-08-07
>
> We greatly appreciate your insight and agree that including an ablation experiment to support this statement is crucial for the integrity and quality of our paper.
>
> ## **Weakness 1:** missing ablation on grouping loss function
> In response to your suggestion, we are currently conducting an ablation study on each term in the grouping loss function in Eq (12). We will report the performance before the end of the discussion period.
>
> ## **Weaknesses 2:** more information about the dataset
> We present the statistics of the dataset in Figure A. in the attached pdf file.
> Initially, we acquired characters with various ages and interaction motion sequences such as  from Character Creator 4 [1]. Subsequently, we composed scenes featuring multiple characters using Omniverse USD [2] Composer. To facilitate the dataset generation process, we engineered the Kaolin rendering tool [3], incorporating tasks such as multi-person normalization to achieve the desired outputs. As a result, we generated multi-view rendered images, mask images, instance masks, and 3D geometry. Furthermore, we created Normal ground truth maps and joints, although these were not utilized in this paper.
>
>
>
> ## **Weakness 3:** SMPL close interactions representation
> > However, I don’t understand why having individual instances of SMPL would complicate the representation of close interactions? I think this is not necessarily true.
>
> In frameworks like DMC that utilize SMPL, the initial SMPL parameters for individual humans are obtained and then optimized to determine the entire scene. In contrast, our approach derives the human pose and shape for the entire scene in a one-shot manner. We intended to mention that using individual instances of SMPL could complicate the representation of close interactions due to the need for separating parameter optimization for each person, which might require complex coordination. In the paper, we will revise to distinguish between SMPL and the models that use it.
> > I agree that it does not naturally include a representation for interactions, but this could be added to the SMPL model. Now saying that it complicates the representation may be too much?
>
> We understand your point about the potential benefits of integrating the SMPL model for multi-person interaction geometry. Applying multi-person interaction geometry to the SMPL model is a reasonable approach that can enhance interaction modeling.
> While existing methods [4,5,6] optimize SMPL pose estimation for multi-person scenarios, a potential limitation is the need for multiple optimization steps. We believe that leveraging a representation for interactions can effectively optimize the SMPL parameters and address these configuration and optimization challenges. This approach can provide a more accurate and computationally efficient solution for modeling close interactions between multiple individuals.
>
>
>
> ## **Weakness 4:** baseline models
> For the baseline models discussed in Section 4.3, we employed different SMPL acquisition methods tailored to each dataset. Specifically, for the HI4D dataset, we used MVPose, aligning with the experimental setup described in the HI4D paper. In their experiments, DMC was run on the HI4D dataset using MVPose for SMPL acquisition, and we adopted this approach to ensure consistency and comparability.
> However, we encountered challenges when using MVPose for the synthetic dataset, as its modules were trained on real data, resulting in less accurate SMPL estimations for synthetic data. To address this, we utilized Learned Vertex Descent (LVD) for the synthetic dataset. LVD is designed to fit SMPL to a 3D human model (3D scan) and has proven to provide more accurate results in this context.
> While LVD is originally intended for single-person scenarios and may be sensitive to occlusion, it was chosen for the synthetic dataset due to its superior performance in fitting SMPL to 3D scans, which is crucial for achieving accurate results in our study. To ensure the robustness of our data and fairness in our experiments, we will report the results trained with MVPose on the synthetic data.
> | Model setting         | CD↓| P2S↓ | NC↑|
> |-----------------------|---|----|----|
> | DMC  (w. syn(MVPose)) | 0.805 | 0.489 | 0.771 |
> | DMC  (w. syn(LVD)) | 0.631 | 0.495 | 0.768 |
> | Ours | 0.406 | 0.329 | 0.892 |
>
> We will revise our paper to include these detailed results and analyses during the rebuttal for providing comprehensive evidence on the proposed framework.
>
> ## **Weaknesses 5:** minor details
> Thank you for your detailed feedback on our manuscript. We acknowledge the points you raised regarding notation, phrasing, and citation formats. We will address them meticulously in our revisions.
>
> If you have any concerns or questions, feel free to leave comments.
>
>
> References
>
> - [1] Character Creator 4, https://www.reallusion.com/character-creator/
>
> - [2] NVIDIA Omniverse USD Composer, https://docs.omniverse.nvidia.com/composer/latest/index.html
>
> - [3] NVIDIA Kaolin, https://github.com/NVIDIAGameWorks/kaolin
>
>
> - [4] Dong, Junting, et al. "Fast and robust multi-person 3d pose estimation from multiple views." Proceedings of the IEEE/CVF conference on computer vision and pattern recognition. 2019.
>
> - [5] Zijian Dong, Jie Song, Xu Chen, Chen Guo, and Otmar Hilliges. Shape-aware multi-person pose estimation from multi-view images. In Proceedings of the IEEE/CVF International Conference on Computer Vision. 2021.
>
> - [6] Yuxiang Zhang, Zhe Li, Liang An, Mengcheng Li, Tao Yu, and Yebin Liu. Lightweight multi-person total motion capture using sparse multi-view cameras. In Proceedings of the 17026 IEEE/CVF International Conference on Computer Vision. 2021.

---

> > ### Comment · Reviewer_Ar19 · 2024-08-13
> > **Feedback**
> >
> > I thank the authors for their responses and the rebuttal document.

---

> > > ### Author Response · Authors · 2024-08-13
> > >
> > > We are glad to hear that your concerns have been resolved. Thank you once again for your valuable comments and positive rating of our work. Your comments have been invaluable in improving our work.
> > >
> > > ### **Weakness 1** missing ablation on grouping loss function
> > > In response to your preliminary review, we have now conducted an ablation study to assess and report the impact of each component within the grouping loss function as defined in Eq. (12). Specifically, the grouping loss function comprises two key terms: the first is the squared distance, and the second is the exponential penalty. The results of the ablation study are presented in the table below.
> > >
> > > | ablation type | squared distance | exponential penalty | CD ↓| P2S↓ | NC ↑ | CP $( \\delta = 0.05 $) ↑ | CP ($ \\delta = 0.075 $) ↑ |
> > > |----|-----------------------|--------------------|---|----|----|---|---|
> > > | (a) | not used | not used  | 0.462 | 0.363 | 0.892 | 0.111 | 0.187 |
> > > | (b) | used       | not used | 0.400 | 0.314 | 0.892 | 0.345 | 0.528 |
> > > | (c) | not used  | used       | 0.532 | 0.403 | 0.880 | 0.228 | 0.335 |
> > > | (d) | used       | used       | 0.406 | 0.329 | 0.892 | 0.629 | 0.670 |
> > >
> > > In the table, model (a) is trained without the grouping loss, while model (d) is trained with the grouping loss.
> > >
> > > It is important to note that the application of the exponential function in the second term encourages soft assignment to a specific instance or cluster. However, using only this term does not lead to improved grouping performance. The combination of both terms within the grouping loss function results in overall performance enhancement.
> > >
> > >
> > > If you have any further questions or additional comments, please feel free to leave a comment.

---

> > > > ### Comment · Reviewer_Ar19 · 2024-08-13
> > > > **Feedback**
> > > >
> > > > I see, thanks for the effort to craft this response, the table looks good and I would encourage the authors to include it on the main text as it can potentially strengthen the paper.

---

> > > > > ### Author Response · Authors · 2024-08-14
> > > > >
> > > > > We will incorporate the table above into the main text, as we agree that it enhances both clarity and the overall quality of the paper. We sincerely appreciate your insightful comments and suggestions throughout the rebuttal and discussion period.

---

### Official Review · Reviewer_A26V · 2024-07-11

**Soundness:** 3
**Presentation:** 3
**Contribution:** 3
**Rating:** 6
**Confidence:** 4

**Summary:**

In this paper, the authors introduce an implicit field representation for multi-person interactive reconstruction. As they said, it can simultaneously reconstruct the occupancy, instance identification (ID) tags, and contact fields. The local-global feature learning methods are used. They also propose a dataset. The result is also good.

**Strengths:**

The idea, motivation, and model are all clearly presented. The method learns features both locally and globally, and I like the projection operation for the query point as shown in equation (3). Also, the design of the loss function is reasonable. The results also demonstrate the effectiveness of the method. The authors propose a new dataset, which is also a significant contribution.

**Weaknesses:**

Since I'm not especially well-versed in this area, I did extensive research on relevant works, such as following [1,2,3] etc. However, I noticed a lack of comparative analysis in the experimental part.  I wonder if this work can be compared with those mentioned? If so, what are the results of such comparisons?
[1]Cha, J., Lee, H., Kim, J., Truong, N. N. B., Yoon, J., & Baek, S. (2024). 3D Reconstruction of Interacting Multi-Person in Clothing from a Single Image. In Proceedings of the IEEE/CVF Winter Conference on Applications of Computer Vision (pp. 5303-5312).
[2]Mustafa, A., Caliskan, A., Agapito, L., & Hilton, A. (2021). Multi-person implicit reconstruction from a single image. In Proceedings of the IEEE/CVF Conference on Computer Vision and Pattern Recognition (pp. 14474-14483).
[3]Correia H A, Brito J H. 3D reconstruction of human bodies from single-view and multi-view images: A systematic review[J]. Computer Methods and Programs in Biomedicine, 2023, 239: 107620.

**Questions:**

This is a quite clear paper. The contribution is also good. I wonder the compare results as said in the weakness. Also I want to know the computational cost, including the training/test time and GPU memory.

**Limitations:**

Yes, the authors discussed the limitations, such as the limitation in finer details.

---

> ### Author Rebuttal · Authors · 2024-08-07
>
> Thank you for your valuable feedback and for recognizing the contributions of our work.
>
> First, we will revise our paper to discuss related works including single-view and multi-view settings mentioned in [1, 2, 3]. We emphasize that the mentioned methods [1, 2] primarily focus on single-view reconstruction not multi-view reconstruction. We designed our overall architecture with multi-view settings in mind from the very beginning to handle occlusion challenges in multi-person interaction cases. A multi-view approach fundamentally incorporates triangulation, making it more robust to unseen scenarios and significantly improving frame-to-frame consistency compared to single-view settings.
>
> Additionally, we understand the importance of providing information on **computational costs**. Here are the details of our computational resources:
> - Training Time: Our model was trained for approximately 2 days with 2 NVIDIA A100 GPUs.
> - Testing Time: Each test instance takes around 60 seconds on the same GPU.
> We will include these computational cost details in the revised manuscript to provide a comprehensive understanding of our approach.
>
> We appreciate your thorough research on relevant works and your interest in comparative analysis. If you have any concerns or questions, feel free to leave comments.
>
> References
>
> - [1] Cha, J., Lee, H., Kim, J., Truong, N. N. B., Yoon, J., & Baek, S. (2024). 3D Reconstruction of Interacting Multi-Person in Clothing from a Single Image. In Proceedings of the IEEE/CVF Winter Conference on Applications of Computer Vision (pp. 5303-5312).
>
> - [2] Mustafa, A., Caliskan, A., Agapito, L., & Hilton, A. (2021). Multi-person implicit reconstruction from a single image. In Proceedings of the IEEE/CVF Conference on Computer Vision and Pattern Recognition (pp. 14474-14483).

---

> > ### Comment · Reviewer_A26V · 2024-08-12
> >
> > Thanks for the responses. All my concerns are addressed.

---

> > > ### Author Response · Authors · 2024-08-13
> > >
> > > We are glad to hear that your concerns have been resolved. Thank you once again for your valuable feedback and positive rating of our work. Your comments have been invaluable in improving our paper. If you have any further questions or additional comments, please feel free to leave a comment.

---

### Author Rebuttal · Authors · 2024-08-07

We sincerely thank the reviewers for their thorough constructive comments on our paper.  We earnestly responded to your concerns; please see the respective comments. Before answering your questions and concerns, we would like to highlight our contributions by quoting your comments.

### **Strengths of the proposed work**
- **Clear presentation**: Multiple reviewers ( **reviewer A26V**, **reviewer Ar19** ) appreciated the clear presentation of ideas, motivations , and models.
- **Innovative ideas and effectiveness**: The innovative methods, such as learning both local and global features and the projection operation for query points, were noted positively (**reviewer A26V**, **reviewer N9qi**). The effectiveness of these methods was demonstrated through strong quantitative and qualitative results (**reviewerA26V**, **reviewer N9qi**).
- **Contribution of new dataset**: The introduction of a new dataset was seen as a contribution by reviewers (**reviewer A26V**, **reviewer Vhig**), indicating its potential usefulness for future research.
### **Weaknesses of the proposed work and raised questions**
- **More information about the dataset** :  In response to **reviewer Ar19** and **reviewer N9qi**, we have provided more comprehensive details regarding the synthetic dataset creation and a detailed explanation of the proposed synthetic dataset and illustrated the data statistics in Figure A. of the attached PDF file.
- **Comparison to SMPL with close interaction**:  **reviewer Ar19** and **reviewer Vhig**  mentioned about SMPL closed interactions representation compared to our method. In frameworks like DMC that utilize SMPL, the initial SMPL parameters for individual humans are obtained and then optimized to determine the entire scene. In contrast, our approach derives the human pose and shape for the entire scene in a one-shot manner.
- **Generalization ability on novel subjects and camera extrinsic parameters**:  These experiments included zoom-in and zoom-out tests with the Hi4D dataset and tests on synthetic data with four individuals performing extreme poses like breakdancing, introducing novel postures and configurations with new camera settings, as shown in Figures C and D of the attached PDF.
- **Computational costs** Requested by **reviewer A26V**, we report the computation costs of our model. We train our model on 2 days with 2 NVIDIA A100 and test 60 seconds for each instance.
- **More experiments**:  **reviewer Ar19** mentioned the need for an ablation study, and **reviewer Vhig** requested a smaller view study. . We are currently training our model for the ablation study and both our model and the baseline model for the smaller view experiments. We will report the results before the end of the author-discussion period.


To accommodate your requests, we have attached an **one-page PDF** for visualization of newly conducted experiments.
If you have any concerns or questions, feel free to leave comments.

---

> ### Author Response · Authors · 2024-08-14
>
> We sincerely appreciate the valuable feedback and discussion throughout the rebuttal and discussion period. Based on comments and discussions, we will revise the current manuscript to strengthen and clarify our framework and explanations. Below, we present additional discussion points addressed during this period.
>
> ### **Additional experiments during discussion period**
>
> As previously mentioned, we have conducted additional experiments with fewer views to address, **reviewer Vhig**'s question about generalization ability and perform an ablation study in response to **reviewer Ar19**'s feedback.
>
> Firstly, our ablation study has yielded promising results, showing that integrating both squared distance and exponential penalty terms into our grouping loss function enhances accuracy and consistency. This outcome not only validates our approach but also directly responds to the feedback provided by **reviewer Ar19**.
>
> | Ablation type | Squared distance | exponential penalty | CD ↓| P2S↓ | NC ↑ | CP $( \\delta = 0.05 $) ↑ | CP ($ \\delta = 0.075 $) ↑ |
> |----|-----------------------|--------------------|---|----|----|---|---|
> | (a) | not used | not used  | 0.462 | 0.363 | 0.892 | 0.111 | 0.187 |
> | (b) | used       | not used | 0.400 | 0.314 | 0.892 | 0.345 | 0.528 |
> | (c) | not used  | used       | 0.532 | 0.403 | 0.880 | 0.228 | 0.335 |
> | (d) | used       | used       | 0.406 | 0.329 | 0.892 | 0.629 | 0.670 |
>
> Additionally, we have addressed **reviewer Vhig**’s concerns about generalization ability by performing experiments with fewer views. Our findings indicate that our method consistently outperforms DMC across all metrics—Chamfer Distance (CD), Point-to-Surface (P2S), and Normal Consistency (NC)—in both 4-view and 8-view settings.
>
>
> | Model (Input views) | CD↓ | P2S ↓| NC↑ |
> |-----------------------|---|----|----|
> | DMC  (4 view) | 1.304 | 0.922 | 0.705 |
> | DMC  (8 view)  | 0.631 | 0.495 | 0.768 |
> | Ours  (4 view) | 0.761 | 0.472 | 0.870 |
> | Ours  (8 view)  | 0.406 | 0.329 | 0.892 |
>
> ### **Explanation on Figures in Attachment**
>
> We missed providing detailed explanations for the meaning of each figure in the attachment during the rebuttal. We would like to address that before the end of the discussion period.
>
> - **Figure A** illustrates the diversity within our proposed dataset, highlighting variations in age, gender, and scene composition. To further enrich this diversity beyond what is seen in Hi4D, we generated synthetic data that includes individuals with varying ages, heights, weights, and garments.
> - **Figure B** displays the instance mesh visualization results produced by our interaction geometry.
> - **Figure C** presents qualitative results from zoom-in and zoom-out experiments, demonstrating that our implicit fields generalize effectively to unseen camera parameters. Our model handles these scenarios well by utilizing a 3D bounding box centered around the individual, preserving the geometry. It particularly excels in zoom-out scenarios for ID field prediction, as it can draw on features from the entire scene.
> - **Figure D** highlights our model’s generalization to new subjects and configurations. The experiments shown in Figure D of the attached document indicate that our model performs robustly with entirely new scene configurations, including new subjects, extreme poses from breakdancing sequences, and new camera settings with varying resolutions and parameters. These results confirm that our model generalizes effectively beyond the synthetic dataset it was trained on.
> - **Figure E** shows adjusted frame examples from our data sample video, ensuring consistent visualization.

---

### Decision · Program_Chairs · 2024-09-25

**Decision:**

Accept (poster)

**Comment:**

Thank you all for the very detailed reviews, rebuttal, and extensive time spent in the discussion. Three out of the four reviewers are positive, with the last one having most of their concerns addressed. The remaining point of reviewer Vhig is the overlap in persons between train/test sets, but this is following the existing protocols/practices. The paper presents a solid approach for representing multi-persion interaction. Please incorporate the additional results and explanation given throughout the rebuttal and discussion phase in the final version.